

# How stratospheric are deep stratospheric intrusions? − LUAMI 2008

Thomas Trickl[1], Hannes Vogelmann[1], Andreas Fix[2], Andreas Schäfler[2], Martin Wirth[2], Bertrand Calpini[3], Gilbert Levrat[3], Gonzague Romanens[3], Arnoud Apituley[4,5], Keith M. Wilson[4,6], Robert Begbie[7], Jens Reichardt[7], Holger Vömel[7,8], Michael Sprenger[9]

5  [1]Karlsruher Institut für Technologie, Institut für Meteorologie und Klimaforschung, IMK-IFU, Kreuzeckbahnstr. 19, 82467 Garmisch-Partenkirchen, Germany
[2]Deutsches Zentrum für Luft- und Raumfahrt, Institut für Physik der Atmosphäre, Münchner Str. 20, 82234 Weßling, Germany
[3]Aerological Station, Federal Office of Meteorology and Climatology, MeteoSwiss, Chemin de l'Aérologie, P.O. 10  Box 316, 1530 Payerne, Switzerland
[4]RIVM, Antonie van Leeuwenhoeklaan 9, 3721 MA Bilthoven, The Netherlands
[5]New address: KNMI, Utrechtseweg 297, 3731 GA De Bilt, The Netherlands
[6]New address: Kipp en Zonen, Delftechpark 36, 2628 XH Delft, The Netherlands
[7]Richard-Aßmann-Observatorium, Deutscher Wetterdienst, Am Observatorium 12, 15848 Tauche, Ortsteil 15  Lindenberg, Germany
[8]New address: NCAR EOL FL-1, 3090 Center Green Drive, Boulder, Colorado 80301, U.S.A.
[9]Eidgenössische Technische Hochschule (ETH) Zürich, Institut für Atmosphäre und Klima, Universitätstraße 16, 8092 Zürich, Switzerland

*Correspondence to:* Dr. Thomas Trickl, thomas.trickl@kit.edu, Tel. +49-8821-183-209, Fax +49-8821-73573

20  **Abstract.** A large-scale comparison of water-vapour vertical-sounding instruments took place over Central Europe on 17 October 2008, during a rather homogeneous deep stratospheric intrusion event (LUAMI, Lindenberg Upper-Air Methods Intercomparison). The measurements were carried out at four observational sites, Payerne (Switzerland), Bilthoven (The Netherlands), Lindenberg (North-East Germany) and the Zugspitze mountain (Garmisch-Partenkichen, German Alps), and by an air-borne water-vapour lidar system creating a transect of humidity profiles between all four stations. A high data quality was verified that strongly underlines the scientific findings. The intrusion layer was very dry with minimum mixing ratios of 0 to 65 ppm on its lower west side, but did not drop below 120 ppm on the higher-lying east side (Lindenberg). The dryness hardens the findings of a preceding study ("Part 1") that, e.g., 73 % of deep intrusions reaching the German Alps and travelling six days and less exhibit minimum mixing ratios of 50 ppm and less. These low values reflect values found in the lowermost stratosphere and indicate very slow mixing with tropospheric air during the downward transport to the lower troposphere. The peak ozone values were around 70 ppb, confirming the idea that intrusion layers depart from the lowermost edge of the stratosphere. The data suggest an increase of ozone from the lower to the higher edge of the intrusion layer. This behaviour is also confirmed by stratospheric aerosol caught in the layer. Both observations are in agreement with the idea that sections of the vertical distributions of these constituents in the source region were transferred to Central Europe without major change. LAGRANTO trajectory calculations demonstrated a rather shallow outflow the stratosphere from just above the dynamical tropopause, for the first time confirming the conclusions in "Part 1" from the Zugspitze CO observations. The trajectories qualitatively explain the temporal evolution of the intrusion layers above the four stations participating in the campaign.

*Key words:* Water vapour, ozone, stratosphere-to-troposphere transport, transport modelling, instrument 40  comparison, lidar, RS92, CFH, LAGRANTO





**1 Introduction**

The complexity of stratospheric air intrusions into the troposphere has been investigated with lidar systems in great detail. A lot of information was obtained from airborne transects (e.g., Browell et al. 1987; 1996; 2001; Flentje et al., 2005) and ground-based time series (e.g., Ancellet et al., 1994; Langford et al., 1996; Eisele et al., 1999; Stohl and Trickl, 1999; Zanis et al., 2003; Trickl et al., 2003; 2010; Di Girolamo et al., 2009; Kuang et al., 2012). Ozone is an excellent tracer for mapping intrusion layers, but does not allow the erosion of these layers within the troposphere to be quantified because one cannot easily resolve the mixing of tropospheric air into the descending layer. Water vapour is a much better choice for such investigations, because of the low stratospheric volume mixing ratio of about 5 ppm (e.g., Scherrer et al., 2008) and only slightly higher values just above the tropopause.

Turbulent mixing has been identified as an important source of tropospheric air in tropopause folds (Shapiro, 1976; 1978, 1980). About half of the air mass in a fold has been estimated to be of tropospheric character (Shapiro, 1980; Vogel et al., 2011). Nevertheless, the tropospheric input has never been quantified along the entire path of the air mass eventually reaching the lower troposphere. An open question has been how much of the tropospheric air originates already from the so-called "mixing layer" around the thermal tropopause (e.g., Danielsen, 1968; Lelieveld et al., 1997; Hintsa et al., 1998; Zahn et al., 1999, 2014; Fischer et al., 2000; Hoor et al., 2002; 2004; Pan et al., 2004; 2007; Brioude et al., 2006; 2008; Sprung and Zahn, 2010; Vogel et al., 2011) prior to the descent and how much of the admixture occurs during the descent of an intrusion layer into the lower troposphere. In some cases mixing of polluted or convectively lifted air into intrusions within the free troposphere has been reported on (e.g., Parrish et al., 2000; Brioude al., 2007; Homeyer et al., 2011; Sullivan et al., 2016).

In contrast to the idea of strong tropospheric mixing Bithell et al. (2000) found in a case study that an extremely dry layer of presumable stratospheric origin survived in the troposphere without resolvable change for at least ten days. Trickl et al. (2014; 2015) verified this behaviour based on a much larger number of water-vapour measurements: In 59 % of the deep intrusion cases with subsidence times up to six days the minimum relative humidity (RH) was 1 % or less, one order of magnitude smaller than the typical results from in-situ measurements with the dew-point-mirror instrument at the nearby Zugspitze summit (2962 m a.s.l.). The corresponding mixing ratio of roughly 50 ppm is typical of values found in the "mixing layer" that extends a few kilometres into the stratosphere.

Despite this evidence of low free-tropospheric mixing the ozone number densities in the same intrusion layers stay significantly below full stratospheric values. Trickl et al. (2014) conclude that the ozone values are mostly determined by how far the intrusion layer initially extends into the stratosphere. They found that CO mixing ratios in deep intrusions rarely strongly differ from tropospheric values. This implies that the descending layers depart from the lowest few kilometres above the dynamical tropopause since fully stratospheric CO values are substantially smaller.

Trickl et al. (2014) discussed three cases with rather filamentary structure in order to demonstrate that exceptionally low mixing prevails even for thin layers. In the follow-up paper presented here, we extend that study by analysing a much more homogenous intrusion layer that was observed over Central Europe during LUAMI (Lindenberg Upper-Air Methods Intercomparison, in the evening of 17 October 2008; Wirth et al.,





2009b). Quantitative three-dimensional mapping with the DLR (Deutsches Zentrum für Luft und Raumfahrt) air-borne lidar system WALES (Wirth et al., 2009a) around a major part of Central Europe are combined with measurements of ground-based lidar systems, balloon-borne sensors at four stations forming the four corners of the flight track and atmospheric transport modelling. The campaign constitutes one of the largest-scale

comparisons of water-vapour profiling instrumentation and verifies a very high quality of all the instruments contributing. In particular, the first comparison of an air-borne and a ground-based differential-absorption (DIAL) system in the entire free troposphere was made. Detailed results are given in the Appendix (Sec. 5).

## 2 Methods

### 2.1 Measurements

#### 2.1.1 DLR Air-borne lidar system WALES

For the validation flight the DLR Falcon F20 aircraft was equipped with WALES, a four-wavelength water-vapour DIAL. The name (Water Vapor Lidar Experiment in Space) was chosen in analogy to the core instrument proposed by DLR for a satellite mission (ESA, 2004). The new instrument design which is described in more technical detail in (Wirth et al., 2009a) features a robust, highly compact and efficient transmitter system which

fulfils all spectral requirements for a water vapor DIAL. The instrument simultaneously emits radiation at three wavelengths resonant with $H_2O$ absorption lines ("on" wavelengths) and at one non-resonant wavelength into the atmosphere ("off" wavelength), Using this set of wavelengths, chosen in a spectral interval between 935 nm and 936 nm, enables to deal with the large dynamic range of water vapor from the planetary boundary layer to the lower stratosphere. The final water-vapour profile is derived as a linear combination of the three profiles

weighted with their reciprocal quadratic uncertainty. Time intervals of 30 s were chosen.

The HITRAN 2008 data base (Rothman et al., 2009) was used as the source of spectroscopic parameters. The high accuracy of the line parameters for the lines selected for the LUAMI flight of 1 to 2 % is verified by the comparisons presented here. A linear combination of the water-vapour profiles retrieved for the three "on" wavelengths, weighted by the squared reciprocal uncertainties is obtained from a statistical analysis of the $H_2O$

profiles.

The density profiles along the flight path were obtained by interpolation of meteorological analysis data (T799L91 resolution; Untch et al., 2006) of the European Centre for Medium-Range Weather Forecasts (ECMWF) for the respective location and time. The T799L91 horizontal grid spacing at mid-latitudes is roughly 25 km, and a 91-level vertical grid up to 0.01 mbar (about 50 levels up to 200 mbar) is used.

WALES provided a transfer standard for comparing the performance of the instruments at the four sites participating in that effort, particularly the lidar systems. The lidar approach makes possible an improved volume matching that is an important prerequisite due to the frequently extreme spatial inhomogeneity of water vapour (Vogelmann et al., 2011; 2015).



### 2.1.2 Payerne instrumentation

The Swiss aerological station Payerne is located approximately 40 km west to south-west of the Swiss capital Bern at 46.8130° N, 6.9437° E and an altitude of 491 m above mean sea level. It is the only permanent Swiss upper-air radiosonde station operated by the Swiss Weather Service, MeteoSwiss, and focuses on the physical

processes and composition of the atmosphere.

Upper air profiles of pressure, temperature, humidity, wind speed and direction are operationally measured at Payerne twice a day, and include three hourly visual weather observations with 24 h staffed operation. Ozone profiles are measured three times per week. In-situ radiosonde profiling has been expanded in recent years with ground-based remote sensing profiling techniques, such as wind profilers, microwave radiometers, a Raman lidar

system and a GNSS (GPS, Global Positioning System) receiving antenna to measure continuously the integrated water vapour column. All surface and remote sensing instruments are in close vicinity to the radiosonde station.

The Raman Lidar for Meteorological Observations (RALMO) is a custom-designed instrument and has been operated at MeteoSwiss Payerne since August 2008. It has been developed by the Swiss Federal Institute of Technology (EPFL) for the needs of MeteoSwiss (for details see (Dinoev et al., 2013; Brocard et al., 2013)).

While other lidar groups (e.g., Leblanc et al., 2008, 2012; Whiteman et al., 2010) have successfully taken the approach of using large integration times during night-time (thus avoiding any daytime sunlight interferences) in order to produce profiles up to the upper troposphere and lower stratosphere, the aim in Payerne is to make continuous measurements of tropospheric water vapor at a high temporal resolution during both day and night. The lidar system uses a frequency-tripled Nd:YAG laser that emits laser pulses (< 8 ns duration) at a repetition

rate of 30 Hz. The typical energy per pulse at the (vacuum) wavelength of 354.8 nm is around 0.3 J, resulting an average power of approximately 9 W. Before being emitted in the atmosphere the beam is expanded to a diameter of 140 mm. This ensures an eye-safe laser beam and reduces beam divergence to 0.1 mrad. Four telescopes with 0.3-m parabolic mirrors are arranged symmetrically around the vertical outgoing beam to receive the backscattered photons. The telescope system has a total aperture equivalent to a telescope of 0.6 m diameter

and a field of view of 0.2 mrad. The narrow field of view together with narrowband spectral filtering in the receiver allows daytime operation. Optical fibres connect the telescope mirrors with a grating polychromator used to isolate the rotational-vibrational Raman signals of nitrogen and water vapor (wavelengths of 386.8 and 407.6 nm, respectively). The optical signals are detected by photomultipliers and acquired by a transient digitizer. The data are stored in half-hour intervals.

An ECC ozone sonde was launched at 13:00 CET (12:00 UTC) together with the operational RS92 radiosonde (Vaisala). Another RS92 sonde was launched at 18:25 CET for the instrument comparison

### 2.1.3 Bilthoven instrumentation

CAELI (CESAR Water Vapour, Aerosol and Cloud Lidar, Apituley, 2009) was set up by RIVM (Rijksinstituut voor Volksgezonheid en Milieu: National Institute for Public Health and the Environment) as a high-

performance, multi-wavelength Raman lidar. The system is meanwhile operated by KNMI (Koninklijk Nederlands Meteorologisch Instituut: Royal Netherlands Meteorological Institute) at the Dutch atmospheric measurement site CESAR (Cabauw Experimental Site for Atmospheric Research, www.cesar-observatory.nl) at





Cabauw (Netherlands). During LUAMI, CAELI was placed at the RIVM compound in Bilthoven (52.12° N, 5.20° E, 5 m a.s.l.). Routine radiosondes are launched from the KNMI station in De Bilt (WMO code 06260, 52.10 N, 5.18 E, 2 m a.s.l.), less than 2 km away from the lidar.

The instrument provides profiles of backscatter and extinction coefficients ($\beta$ and $\alpha$, respectively), depolarisation
and water vapour. Data is collected, suitable as input for retrieving aerosol micro-physical parameters based on so-called $3\beta+2\alpha$ schemes, i.e., based on three backscatter and two extinction channels. Tropospheric coverage is provided, including the boundary layer. Round-the-clock measurements are possible, including a good daytime performance for the UV $N_2$ Raman channel. A 0.57-m diameter far-field receiver doubles the six near field detection channels from the 0.15-m near-field telescope. A third telescope with 2" diameter is used for
polarisation detection at 532 nm. For the water vapour measurements presented in this paper, the relevant emission wavelength of the laser is 355 nm. CAELI nominally emits 10 W at that wavelength. The system is field deployed in a 20-ft sea container for making it transportable. The instrument itself, including the electronics is mounted in a single rugged aluminium frame that can be wheeled in and out of the container in its entirety. Windows are mounted on the top of the frame, above the receivers and in ports for the laser beams, to weather-
proof the system and to avoid beam steering problems due to turbulence above the exit.

The lidar data is ingested at 10 seconds time resolution and 7.5-m vertical sampling. The water-vapour profiles are averaged over 15 minutes, one of them coinciding with the Falcon overpass on 17 October 2008. The water vapour mixing ratio is calculated from the ratio of the 407 nm and 387 nm signals and calibrated against the noon radiosonde at De Bilt. Smoothing is applied to the profile with a range-dependent smoothing length going
from high resolution at low altitudes and progressively lower resolution to the far range.

### 2.1.4 Lindenberg instrumentation

The water-vapour Raman lidar RAMSES (Raman lidar for atmospheric moisture sensing, Reichardt (2012; 2014), Reichardt et al. (2012; 2014)) was installed at the Richard Aßmann Observatory of the German Meteorological Service in Lindenberg (east of Berlin) in 2005 (52° 12' 31.9" N und 14° 07' 18.8" E). It is housed
in a standard air-conditioned 20-foot container. An injection-seeded frequency-tripled Nd:YAG laser serves as the radiation source. Only third-harmonic radiation at 354.84 nm is emitted into the atmosphere. Output power is 15 W at 30 Hz pulse repetition rate. RAMSES is operated with two receiver telescopes simultaneously. A Cassegrain telescope with 800 mm diameter is non-fibre-coupled to the far-field receiver, while a 200-mm Newtonian telescope is fibre-coupled to the near-field receiver. During LUAMI, the detection sections of both
receivers were nearly identical. After the beam collimation, dichroic beam splitters and interference filters separate the elastically backscattered light (354.84 nm) and the ro-vibrational Raman signals of water vapour (407.6 nm) and of molecular nitrogen (386.8 nm). All optical signals are recorded with selected photomultiplier tubes. Data acquisition is performed with Licel analogue and photon-counting transient recorder system. Measurements were performed only during night-time, measurement products were water vapour mixing ratio,
and particle backscatter and extinction coefficients (Engelbart et al., 2006). During the campaign data were prepared as 10-min and 30-min averages. Normally, we present the 30-min data here. For the comparison with the airborne DIAL, both averaging times were taken.



At Lindenberg water vapour profiles are also measured using balloon-borne in-situ sensors. Four times daily balloon launches with Vaisala RS92 radiosonde take place as well as twice monthly addition ally with cryogenic frost-point hygrometers (CFH, Vömel et al., 2007a). Lindenberg is the Lead Center for the GCOS Reference Upper-Air Network (GRUAN) of the World Meteorological Organization. All radiosonde data are processed

with special GRUAN algorithms developed there (Immler et al., 2010; Dirksen et al., 2014). In addition to the routine ascents a dedicated balloon with an RS92, CFH and an EnSci ECC ozone sonde was launched during the campaign to coincide with the Falcon overflight and the horizontal flight path of the aircraft was chosen to match the trajectory of the balloon (upper Panel of Fig. 2).

### 2.1.5 Zugspitze lidar system and in-situ data

The Zugspitze water-vapour DIAL is operated at the Schneefernerhaus high-altitude research station (UFS, 47° 25′ 00″ N, 10° 58′ 46″ E) at 2675 m a.s.l., about 8.5 km to the south-west of IMK-IFU (Garmisch-Partenkirchen, Germany), and 0.5 km to the south-west of the Zugspitze summit (2962 m a.s.l.). The full details of this lidar system were described by Vogelmann and Trickl (2008). It is based on a powerful tunable narrow-band Ti:sapphire laser system with up to 250 mJ energy per pulse operated at about 817 nm (here: about 150 mJ) and

a 0.65-m-diameter Newtonian receiver. Due to these specifications a vertical range up to about 12 km is reached, almost independent on the time during the day. A separation of near-field and far-field signals is achieved by a combination of a beam splitter and a blade in the far-field channel. In this way the operating range starts below the altitude of the summit station (2962 m a.s.l.). The vertical resolution chosen in the data evaluation is dynamically varied between 50 m in altitude regions with good signal-to-noise ratio and roughly 350 m in the

upper troposphere. Free-tropospheric measurements during dry conditions clearly benefit from the elevated site outside or just below the edge of the moist Alpine boundary layer (e.g., Carnuth and Trickl, 2000; Carnuth et al., 2002). After a few years of testing, validating and optimizing the system routine measurements were started in January 2007 with typically two measurement days per week, provided that the weather conditions are favourable.

On the basis of the comparison with the DLR DIAL a minor deficiency in the calculation of the spectral line wings could be detected and was corrected. The choice of spectral line parameters (Ponsardin and Browell, 1997) is justified by the excellent results (Sec. 3.6). A more recent comparison with the Zugspitze Fourier transform spectrometer confirmed this performance and revealed slight discrepancies for some 817-nm lines taken from the HITRAN (Rothman et al., 2009) data base (Vogelmann et al. 2011). Furthermore, in that study, a

very high importance of volume matching in comparisons of water-vapour profiling instruments was found (see also (Vogelmann, 2014)).

In addition, in-situ data from the monitoring station at the Zugspitze summit are used, namely ozone, carbon monoxide, and relative humidity. Ozone has been measured since 1978 (e.g., Reiter et al., 1987; Scheel et al., 1997; Oltmans et al., 2006; 2012; Logan et al., 2012; Parrish et al., 2012). Recently, ultraviolet absorption

instruments have been employed (TE49 analysers, Thermoelectron, U.S.A.). Carbon monoxide was measured using vacuum resonance fluorescence (AL5001, AeroLaser, Germany). RH was registered with a dew-point mirror (Thygan VTP6, Meteolabor, Switzerland) with a quoted uncertainty below 5 % RH. However, the instrument has a wet bias of almost 10 % under very dry conditions (Trickl et al., 2014).



The tropospheric ozone lidar at Garmisch-Partenkirchen, Germany (IMK-IFU; 47º 28′ 37″ N, 11º 3′ 52″ E, 740 m a.s.l.; Kempfer et al., 1994; Carnuth et al., 2002; Eisele and Trickl, 1997; 2005) was out of operation because a computer failure occurred during the warm-up for the campaign.

### 2.2 LAGRANTO Model

Five-day forward trajectories are calculated for the time period from 1:00 CET (Central European Time, = UTC + 1 h) on Oct 8, 2008, until 19:00 CET on Oct 15, 2010 every six hours based on the Lagrangian Analysis Tool (LAGRANTO; Wernli and Davies, 1997; Sprenger and Wernli, 2015). The three-dimensional wind fields for the calculation of the trajectories were taken from ERA-Interim data set (Dee et al., 2011) from the European Centre for Medium-Range Weather Forecasts (ECMWF), which was interpolated to a longitude-latitude grid with 1×1

horizontal resolution and provides six-hourly winds.

The large set of five-day trajectories was started in the entire region covering the Atlantic Ocean and Western Europe (20º E to 120º W and 40 to 80º N) between 200 and 600 hPa. Then, from this set of trajectories those initially residing in the stratosphere (potential vorticity (PV) larger than 2.0 pvu) and descending during the following five days by more than 300 hPa into the troposphere were selected as "stratospheric intrusion

trajectories". The same PV and pressure selection criteria were used in a previous case study (Wernli, 1997) to study an intrusion associated with a major North Atlantic cyclone, in daily operational intrusion forecasts for the former STACCATO (Stohl et al., 2003) observational network (Zanis et al., 2003; Trickl et al., 2010; maintained to the present day), and was also used with in a recent study about stratospheric intrusions (Trickl et al., 2014).

### 3 Results

### 3.1 Characterization of the Stratospheric Air Intrusion on 17 October 2008 by Trajectories

The intrusion was first detected in the routine forecast plot daily sent to former STACCATO (Stohl et al., 2003) partner stations (Zanis et al., 2003). Here, we give in Fig. 1 a revised version of that plot, now based on ECMWF re-analysis meteorological data, and based on the all trajectories calculated for the period between 8 October 2008, and 19:00 CET  on 15 October 2008, fulfilling the criteria for deep stratosphere-to-troposphere transport

(STT) specified in Sec. 2.2. From these trajectories, Fig. 1 shows those intersecting the 65º meridian between 60º N and 75º N within six hours from 15 October, 1:00 CET. This time was chosen for a maximum horizontal extension of the trajectory field towards Payerne that occurred during the second half of 17 October when the measurements took place. The air pressure range of the main layer over Central Europe is roughly 650 mbar to 750 mbar (about 2.5 km to 3.7 km), the lower altitudes being found to the south-west, the higher ones to the

north-west of the trajectory bundle, in agreement with typical behaviour.

### 3.2 DLR Measurements

The flight path of the DLR Falcon jet is marked in Fig. 1. Colour-coded summary plots of the measurements during the flight on 17 October 2008 are given in the panels of Fig. 2. The flight started at Oberpfaffenhofen (ICAO (International Civil Aviation Organization) code EDMO) at 16:42 CET. The aircraft turned to the west

and climbed to about 11 km altitude. It first reached Payerne at 17:18 CET, then Bilthoven at 18:15 CET, Lindenberg at 19:03 CET, and finally Garmisch-Partenkirchen (Zugspitze) at 19:52 CET The data are most





accurate in the upper troposphere, i.e., close to the aircraft, but are remarkably reliable even in the lower troposphere, where the lidar signal is much weaker and, thus, noisier. The relative noise level within the dry layer additionally grew whenever the water-vapour density above the intrusion was enhanced to an extent that much of the radiation was absorbed. The lower-tropospheric performance was, thus, the best over Lindenberg
(see Fig. 2) and becomes evident from the comparisons that are shown in the Appendix, with one exception.

In the lower panel of Fig. 2 also the backscatter ratio for 1064 nm is given, i.e., the ratio of the total backscatter coefficient and the Rayleigh backscatter coefficient. Any value exceeding 1.0 means the presence of aerosol, and high values around the upper end of the scale can be attributed to clouds. The data gaps (white areas) are mostly associated with the presence of clouds at the top of the boundary layer or cirrus clouds and the corresponding
light loss.

Quite importantly, slightly enhanced aerosol was retrieved in the upper half of the intrusion layer. The most reasonable explanation of this observation would be a downward transport of some of the enhanced stratospheric aerosol after the violent eruptions of Okmok and Kasatochi (to 15 km and 13.7 km, respectively (Massie, 2015)) around the middle of 2008 that was also registered with the stratospheric aerosol lidar at Garmisch-Partenkirchen
(Trickl et al., 2013), up to about 19 km in October 2008.

### 3.3 Payerne

During the hours of the LUAMI campaign Payerne was located close to the western edge of the intrusion layer (Fig. 1). Nevertheless, the time series of the Raman lidar (Fig. 3, upper panel) verifies the presence of a very dry layer between 2 and 3 km during the entire period displayed, starting at 13:45 CET. The driest period with
mixing ratios of 35 to 65 ppm started at about 17:30 CET (Fig. 3, lower panel). 50 ppm is a typical value as found in the tropopause region (Trickl et al., 2014). The relative uncertainties of the minimum mixing ratios specified for the period before 17:00 CET are 7 to 19 %, after 17:00 CET 5 to 9 %.

The presence of stratospheric air is confirmed by the 13:00-CET ozone profile (Fig. 4) that exhibits a 76.2-ppb maximum at 3.2 km, residing on a background of roughly 50 ppb. It is interesting to note that the corresponding
RH minimum is downward shifted by about 0.3 km. The midnight (0:00 UTC or 1:00 CET) RH minimum was 1 %, presumably a truncation value (Trickl et al., 2014). This low value is in agreement with the drier situation revealed by RALMO for the night.

### 3.4 Bilthoven

The time series of the CAELI system is depicted in Fig. 5. The noise at early times is due to clouds passing over
the lidar. Two dry layers are visible. However, the minimum mixing ratios are of the order of 500 ppm (Fig. 6) which is beyond typical values in the lowermost stratosphere. By contrast, the noon sonde measurement at De Bilt (KNMI) (Fig. 6), as in the case of Payerne, shows the typical low-humidity cut-off at 1 % RH (about 70 ppm). Even 70 ppm are, again, within the range of values frequently found just above the tropopause. It seems that that at the time of the lidar measurements in Fig. 5 the driest part of the intrusion was already over. Around
midnight, the intrusion layer had almost disappeared as can be concluded from the 24:30 CET sonde measurement.





The two lidar systems agree well in a range up to 8 km (Fig. 6). There are just a few exceptions outside the specified uncertainties most likely due to insufficient spatial matching, or far-field detection of the DLR lidar. The agreement with the sonde data is not satisfactory due to the considerable time differences, except for the range between 3.2 and 7 km in the midnight profile.

In addition, a profile from the ECMWF analysis is shown. Outside the dry layers the agreement is reasonable, but just one of the two layers seen in the measurements is indicated. Another example can be found in the Appendix (Payerne, Fig. A1).

**3.5 Lindenberg**

During the campaign the lidar data of RAMSES were prepared as 10-min and 30-min averages. The time series
of the 30-min data is shown in the upper panel of Fig. 7, also using some 10-min data next to the data gaps. The measurements were continued until 6:00 CET on 18 October. During the period displayed the intrusion layer became continually thinner. The data do not exhibit a single minimum of the mixing. In the lower panel of Fig. 7, we, therefore, show the minimum values for the two driest zones in the upper panel separately. The minimum mixing ratios retrieved are 120 ppm, which is, still, in some agreement with UTLS conditions inside the "mixing
layer", but clearly higher than the minima observed at the other sites. The relative uncertainties of the RAMSES mixing ratios specified in the vertical range around the intrusion are just a few per cent.

The ozone profile measured by the balloon payload launched at 18:44 CET is shown in Fig. 8. Quite interestingly, the highest ozone peak (75 ppb) was observed at the upper end of the dry layer at an altitude of about 5.5 km, although just 0.1 km above the RH minimum (5 %). This is in agreement with the idea that the
ozone rise in the lowermost stratosphere of the Arctic source region was transferred to Lindenberg without major change, assuming low interference by tropospheric air during the transport (Trickl et al., 2014). A similar behaviour is indicated for Payerne in Fig. 4.

The ozone structure above 6 km is not clear. There is an obvious anti-correlation of ozone and RH indicating stratospheric influence. However, the elevated RH values could indicate mixing with tropospheric air.

In Fig. 9 a colour-coded plot of the water-vapour mixing ratio derived from the radiosonde ascents at Lindenberg between 14 and 23 October is shown. The plot benefits from the six-hour intervals between the launches at Lindenberg, shorter than the conventional 12 h. If one neglects uncertainties due to the graphical procedure applied, there is a strong hint on a direct connection of the dry layer to the stratosphere during the first half of 17 October (Julian day 291) that is also indicated in the upper panel of Fig. 7. The transverse drift of the fold away
from Lindenberg is confirmed by the model calculations (Sec. 3.7).

The aerosol backscatter coefficients derived from the 354.84-nm RAMSES measurements are rather noisy due to the very strong contribution from Rayleigh backscattering at this short wavelength. Nevertheless, a small spike (backscatter ratio 1.05) is seen in the profile next to the DLR overflight at 5.08 km (not shown), residing on a broader pedestal between 3.8 and 5.2 km. This structure is in good agreement with the WALES results (Fig. 2).
The result of a 3-h average is shown further below (Sec. 3.6).





### 3.6 Zugspitze

On 17 October 2008 a total of five measurements with the water-vapour DIAL at UFS were made between 16:55 and 20:55 CET. Figure 10 gives an overview of the profiles. The data are given as number densities which is the primary quantity measured by DIAL systems (not requiring the additional use of sonde data). During that time period the intrusion layer descended by about 0.6 km. The minimum densities ranged between $-7.7 \times 10^{19}$ m$^{-3}$ (the negative value being caused by data noise) and $7.9 \times 10^{20}$ m$^{-3}$ m$^{-3}$, with a standard deviation of $7 \times 10^{20}$ m$^{-3}$ (corresponding to a mixing ratio of roughly 37 ppm). The figure suggests that the intrusion cut a descending dry hole into a triangular humid distribution that was gradually restored, as indicated by the growing peak density.

The noon and midnight RH profiles of the Munich (Oberschleißheim, WMO station 10868, 100 km roughly to the north) sonde type RS92) extend the range of descent over southern Bavaria to 3.9 km (thick red arrow in Fig. 10) → 2.78 km. Because of the complexity of Fig. 10 we do not include the corresponding H$_2$O density profiles there. We used high-resolution data received from the German Weather Service (DWD). In this data set, all four minima for Munich and Stuttgart between noon and midnight reach the cut-off value of 1 % RH. The situation seems to differ from that in Lindenberg: the tropopause for the preceding ascent (1 CET on 17 October) does not exhibit a strong lowering and the RH values are rather high throughout the troposphere. However, the time difference of 12 h is too long to be absolutely sure about excluding a direct connection of the dry layer to the stratosphere over South Germany.

It is interesting to note that, despite uncertainties of the sonde results, the value of 1 % RH has been found to be quite typical in the routine analyses of STT events at Garmisch-Partenkirchen since 2007. This value is clearly dominating for low to moderate travel times. For subsidence times beyond ten days the RH minima may grow to 2-6 %. In the current study 1 % RH was consistently observed in the sonde data in the vicinity of all sites involved but Lindenberg, where also the lidar minima are slightly higher.

The Zugspitze in-situ measurements showed a drop in relative humidity right after the end of the lidar comparison in agreement with further descent of the dry layer (Fig. 11). The minimum half-hour average, 7.2 %, was not reached before 1:00 CET, which indicates considerable slowing of the subsidence. A pronounced ozone rise to more than 73.3 ppb was found that started four hours later than the beginning of the humidity drop. Both the peak ozone value and the delay are in agreement with the findings for Payerne and Lindenberg. Carbon monoxide did not change much, which is rather typical (Trickl et al. (2014). Fully stratospheric CO values are substantially lower.

As in the 1064-nm measurements of WALES aerosol the UFS measurements show aerosol in the upper half of the intrusion layer, with a peak 817.2-nm backscatter coefficient of about $1.47 \times 10^{-7}$ m$^{-1}$ sr$^{-1}$ (Fig. 12; the corresponding backscatter ratio is 1.76). The aerosol structure could be so clearly detected by both DIAL systems because of the low noise of the rather small Rayleigh background at the long wavelengths used. We show two examples, one profile in the late afternoon (17:02 CET) that contains the entire aerosol peak centred at 3.57 km, but ending below a cirrus layer, and another profile around the time of the comparison when the aerosol peak was located at the lower edge of the useful range of the backscatter profile. The second profile was also evaluated in the stratosphere and shows two peaks of the volcanic eruptions (at about 12.7 km and 16.2 km), in addition to the stratospheric background that extended from the tropopause to about 25 km during the



background phase preceding the eruptions (Trickl et al., 2013). The stratospheric peaks are considerably smaller than the peak inside the intrusion layer because of the much lower atmospheric density. This explanation assumes that a comparable aerosol density was also present over the source region, which looks reasonable many months after the eruptions. However, full homogeneity was not reached as seen in the figure and as was

discernible in the aerosol profiles of the DIAL that varied from hour to hour on that day. In the lower panel of Fig. 12 an expanded section of the profile for 17:02 CET is given, together with the corresponding water-vapour profile rescaled to fit into the plot window. Clearly, the aerosol peak is located in the upper half of the intrusion layer, as already concluded from Fig. 2. This resembles the behaviour of the ozone distribution at Payerne and Lindenberg.

In addition, the 3-h average for Lindenberg around the overflight time is inserted into the upper panel in grey colour. The curve is rescaled by multiplying the values with $354.84/817.2$ according to a $\lambda^{-1.4}$ Ångström law guessed from the curves of Jäger and Deshler (2002) for the wavelength dependence of the backscatter coefficients. The Lindenberg aerosol peak inside the intrusion is located at 4.97 km ($2.3 \times 10^{-7}$ m$^{-1}$ sr$^{-1}$, backscatter ratio 1.05), i.e., slightly downward shifted due to the long averaging. The tropopause above

Lindenberg was at just 10.1 km, which explains the lower position of the lower volcanic layer just above this altitude.

### 3.7 Transport modelling

The five-day trajectories were released and preselected for deep subsidence from the lowermost stratosphere as described in Sec. 2.2. In the next step, cross sections transverse to the flow were prepared at a number of

locations between Canada and the Alps. Examples for four of the locations are shown here. The PV contours (in colour), isentropes (as blue contour lines), the interpolation points of the individual trajectories closest in time (within ±6 h) to the cross sections (yellow dots) and the points of intersection of the trajectories (magenta dots) are displayed. The cross sections allow the position of the air parcel to be seen relative to the dynamic tropopause (2-pvu isosurface) and highlight their way down from the stratosphere to the lower troposphere. Note

that all trajectories calculated, i.e., starting between 8 and 15 October and fulfilling the deep-intrusion criteria, contribute to the cross sections.

The temporal development of the tropopause and positions of the trajectories as they cross a first vertical surface (along 80° W, green line) north of Hudson´s Bay are shown in Fig. 13. The figure shows three examples selected from the full time period 14 October, 6 UTC, to 15 October, 12 UTC, representing the phase of the highest

trajectory density at this cross section and the first indication of diminishing. The trajectories (black lines) are shown in the right panels.

At the location of the transverse surface of Fig. 13, the tropopause is only slightly distorted toward lower altitudes, during the entire period covered. The beginning of the trajectories selected by the deep-STT criterion stays east of 100° W, i.e., well inside the model domain. This means that the chosen control surface is close to

the true beginning of the intrusion. The tropopause is located clearly below the minimum pressure level of 200 mbar. Please, note that the trajectories concentrate not far from the 2-pvu surface. This nicely confirms the conclusion of Trickl et al. (2014) from the rather high Zugspitze CO values in intrusions (see also Fig. 11) that the intrusions emerge from a shallow layer just above the dynamical tropopause.





In the next cross section farther downstream (50º W, Fig. 14) already a fully developed tropopause fold is seen. The highest trajectory density in the vicinity of this cross section was found between 6:00 UTC and 18 UTC on 15 October. Here, we select the situation for 12 UTC as an example for which also the lowest position (about 520 mbar) of the 2-pvu contour and the most pronounced westward extension of the intrusion over Central

Europe (corresponding to the most pronounced dryness over Payerne) were obtained. It is interesting to note that for the entire 12-h period of maximum stratospheric density the trajectories intersect the control surface above the centre of the fold. The lowest deviation from the centre was found for 6 UTC.

The best coincidence with the next transverse surface at 30º W (Western Iceland, not shown) was calculated for the period 18 UTC to 24 UTC on 15 October, the lowest position of the dynamical tropopause (about 500 mbar)

occurring at 6 UTC on 16 October. However, at this time the trajectory dots were positioned even fully above the fold.

The next surface was selected from 50º N, 0º E to 54º N, 20º E, approximately representing Bilthoven and Lindenberg (shifted less than 1º to the north). In Fig. 15 we show the panels for 12 UTC to 24 UTC on 17 October. The trajectories cover Bilthoven in the first two right panels, but move eastward towards midnight. This

is in qualitative agreement (though slightly later) with the rising minimum humidity in the observations. Over Lindenberg, the trajectories seem to confirm the extended vertical range (roughly 800 to 600 mbar) seen in the lidar measurements. In addition, the sequence of panels shows an eastward propagation of the fold along the control surface, in agreement with the radiosonde measurements shown in Fig. 9.

The trajectories in Fig. 15 pass east of Payerne. Those covering Payerne reach the coastal area six to eighteen

hours earlier (not shown).

Finally, a cross section slightly north of the Alps (44.5º N, 2º E to 51º N, 18º E) was prepared (Fig. 16), almost exactly hitting Payerne and passing 0.9º north of Garmisch-Partenkirchen/Zugspitze. We cut off the cross section to the north east, not reaching the end of the intrusion. This decision was made because a number of stratospheric trajectories from outside the trajectory field in Fig. 1 (not intersecting the first control surface at 80º W) is

located there and adds complexity. In any case, the radiosonde measurements at Payerne and Munich do not indicate a connection of the intrusion layer to the stratosphere as in the case of Fig. 9. However, the radiosonde ascents at these stations took place at longer intervals of 12 h, perhaps too coarse to see more details.

In Fig. 16 we give three examples of model calculations again for 12 UTC to 24 UTC on 17 October. During this time the best overlap of the trajectories with Payerne is found, in agreement with the growing dryness observed

during this period. Due to the cut-off towards the north –east mentioned above the trajectory dots do not reach the high-PV contours which is the case for a longer control surface.

It is obvious that the trajectory dots in the cross sections downstream the intrusion exhibit a higher spread. To some extent this is ascribed to the higher temporal jitter and to additional stratospheric contributions from outside the main descending air stream. The best overlap with the contour of the fold is seen for the later times,

in agreement with the driest phase observed over Payerne, although the trajectories for the beginning of 18 October no longer overlap with the Swiss station.



**4 Discussion and Conclusions**

There is growing evidence that ozone injection from the stratosphere is very likely a much stronger source of tropospheric ozone than frequently thought (e.g., Roelofs and Lelieveld, 1997; Trickl et al., 2010; 2011; 2014). However, a quantification of STT remains a difficult task. The results presented in this paper, together with the findings of the preceding studies (Trickl et al., 2014; 2015), are an important prerequisite on the way to quantifying STT based on observational data alone, at least at a few suitable stations: The low concentrations of water vapour found in deep stratospheric intrusions suggest that the intrusion layers reach high-lying atmospheric observatories with rather little modification during the transport. The long-term observations of ozone, RH and [7]Be at these stations can, therefore, yield a reasonable estimate of the impact of STT at these sites.

The LUAMI measurements on 17 October 2008 have made possible a thorough comparison of different high-quality instruments for water-vapour sounding, in particular the CFH sonde, differential-absorption and Raman lidar systems. The air-borne lidar served as a transfer standard. With respect to the intercomparison of the instruments, the following main conclusions can be drawn:

- Apart from a generally excellent mutual agreement of the systems a high capability of determining very low humidity levels was verified such as those needed in the current study. The RS92 radiosonde (e.g., Miloshevich et al., 2006; Vömel et al., 2007b; Steinbrecht et al., 2008; Dirksen et al., 2014)) was verified to reproduce RH values around 1 % indicating a capability of resolving even lower values. The ground-based lidar systems were found to resolve significantly lower humidity in the deep stratospheric air intrusions since these layers are measured at relatively short distances.

- The campaign was to a major extent based on lidar measurements. Lidars are ideal due to the important (Vogelmann et al., 2011; 2014) advantage of volume matching and of producing dense time series. Since the air-borne DIAL provided information of the spatial structure of water vapour also the quality of the balloon-borne instruments could be judged. At Lindenberg the spatial matching of WALES and the balloon was particularly good since the aircraft flew along the wind direction.

- The signal of Raman lidar systems (Payerne, Bilthoven and Lindenberg) is proportional to the $H_2O$ density divided by the square of distance. As a consequence, these systems are advantageous for measurements under very dry conditions, at least during night-time. Without a noisy solar background the humidity determined from Raman lidar systems is positive from its very principle since each signal photon is caused by backscattering of the laser radiation by $H_2O$. At least after 18:00 CET on 17 October the measurements of the Raman lidar systems are invaluable for this study since they yield very reliable values for the humidity minima in the intrusion layer above the respective site (uncertainty: roughly 5 ppm $H_2O$).

- The water–vapour data of DIAL systems in dry layers are noisy at all times since they are based on absorption measurements in a noisy backscatter signal. Under very dry conditions the noise can lead to pointwise negative humidity values. However, as concluded previously (Trickl et al., 2014), the comparisons confirmed that also DIAL systems can rather reliably determine low values: As found by Trickl et al. (2014), the minimum uncertainty of the ground-based Zugspitze DIAL in dry layers in the lower



free troposphere under optimum conditions is of the order of 25 ppm (roughly $5\times10^{20}$ m$^{-3}$ as to density, or 0.5 % RH).

With respect to the dynamics of the intrusion some interesting findings could be found based on the measurements and the modelling study. In particular, the observations, carried out in a rather wide region,

together with the model calculations have led to a thorough characterization of the intrusion. The cross sections prepared with LAGRANTO trajectories nicely show the development of the main intrusion layer from the source region in arctic Canada on its way to the Alps. The main conclusions are:

- The minimum water-vapour mixing ratios observed above most sites participating were clearly below 100 ppm during the driest periods, the lowest values having been about 50 ppm or less (Payerne and Zugspitze).

The dryness above Payerne is a remarkable fact since this station was close to westernmost edge of the intrusion. The low values harden the conclusions of Trickl et al. (2014) that significant mixing of the stratospheric air during the downward transport to 3-4 km takes only place if there is external interference from nearby frontal systems or convection. Stratospheric air layers can travel over very long distances without losing much of their characteristics (Trickl et al., 2014, 2015). Sometimes they survive with minor

mixing even when travelling once around the globe (Trickl et al., 2011).

- The formation of the tropopause fold starts significantly earlier and at slightly higher altitudes than anticipated from the daily forecasts received since autumn 2000 (Zanis et al., 2003). However, the success of the forecasts (Trickl et al., 2010) could be due to the the fact that a minimum start pressure of 250 mbar (about 10.5 km) stays within the range of lowered tropopause positions in the fold region, even in summer.

- The trajectory bundles transversely propagating in the folds initially stay rather narrow, narrower than the fold structure marked by the PV = 2 pvu contours. Later on, the bundle seems to expand, to some extent due to the temporal spread at the control surfaces and due to additional STT contributions entering the cross sections. Acute-angled fold structures have repeatedly been observed with the ozone DIAL over Garmisch-Partenkirchen (e.g., Trickl et al., 2010).

- Downward motion occurs all along the path, initially faster on the west side.

- The changes in trajectory density and position shifts of the trajectory bundle qualitatively confirm the time periods of the driest parts of the layer in the observations at the different sites.

The LUAMI measurements of water vapour, ozone and aerosol have indicated another behaviour of descending stratospheric layers. As hypothesized by Trickl et al. (2014) the ozone and aerosol distributions in the intrusion

layer is in agreement with the idea of an rather unperturbed transfer of the distribution of these species in the source region to Europe: An increase of ozone towards the top of the intrusion was documented at three stations, the location of the aerosol peak in the upper part for the entire DLR flight. The straight air flow out of the lower stratosphere revealed by the model calculations (transverse to the fold) confirms this idea. More cases must be analysed to harden these findings.

As in the vast majority of the ozone observations with the lidar at Garmisch-Partenkirchen the peak O$_3$ mixing ratio stays moderate. Most intrusions originate in the lowest layer above the dynamical tropopause as concluded by Trickl et al. (2014) from the almost negligible drop in Zugspitze CO that is reproduced in Fig. 11. This is now clearly verified by the modelling results (Fig. 13). Exceptions are rare. For example, on 1 October 2015 a layer almost 7 km wide with up to 235 ppb of ozone was registered with the ozone lidar, between 5 and 11 km a.s.l.





However, this is, still, far away from peak ozone mixing ratios of the order of 5 ppm found in the stratosphere above 20 km.

The aerosol seen in Figs. 2 and 12 in the upper half of the dry layer seems to reflect the behaviour of ozone, which increased backscatter coefficients towards the layer top. It is reasonable to assume that the lower volcanic peak was located just above the tropopause in a major part of the northern hemisphere. Stratospheric aerosol in intrusion layers has been rarely reported (e.g., Browell et al., 1987). We have seen indications in the ozone plus aerosol soundings at Garmisch-Partenkirchen in 2009 following the Sarychev eruption, or after a 1991 pyro-cumulonimbus in the Québec province of Canada (Carnuth et al., 2002; Fromm et al., 2010). STT has been seen as the most important mechanism in the mid-latitudes, limiting the stratospheric dwell time of aerosol in the mid-latitude stratosphere to one year and less (Trickl et al., 2013). As a consequence, also the stratospheric impact of boreal smoke plumes (e.g., Fromm et al., 2008, and Fig. 1 of Trickl et al. (2013)) or particle formation from aircraft emissions at high cruising altitudes is lowered.

In Figs. 6 and A1 (see below) high-resolution ECMWF profiles are presented. These profiles were calculated for the entire flight track (Wirth et al., 2009). The ECMWF analysis shows roughly the same $H_2O$ distribution as the measurements, but (as in Figs. 6 and A1) it is apparent from its much smoother structure that the model is by no means able to resolve the fine structure of the dry layer. The mean deviation between the WALES measurements and the ECMWF analyses is −13% (i.e., WALES is dryer). If the altitude region of the dry layer is excluded, the mean difference is about −8%.

**2 2 Appendix: Instrument Comparison**

We present in the following the results of the instrument comparisons that are not primarily relevant to the scientific discussion of this paper.

*Payerne*

The DLR Falcon passed over Payerne at 17:18 CET, i.e., during the transition period towards the lowest water-vapour mixing ratios (Fig. 4). The comparison of the two lidar systems and the profile obtained from an extra RS92 ascent is shown in Fig. A1. The minimum mixing ratios from the DLR DIAL and from the sonde agree well, whereas the minimum for the Raman lidar is slightly higher. This deviation is outside the uncertainty specified for RALMO (12 ppm), but inside that of the WALES data (as high as 200 ppm due to the strong radiation loss in the moist layer above the intrusion). The RALMO profiles are not shown beyond 5 km due to a deteriorating performance caused by the background noise from residual daylight. For comparison, the uncertainty at the humidity minimum during the dark phase was 3 ppm.

In addition, a profile of the water-vapour mixing ratio from the ECWMF T7699L91 analyses is given (See Sec. 2.1.1.). The agreement outside the intrusion is rather good, but the intrusion is not only strongly underestimated, and it is also vertically shifted in the model output.

*Lindenberg*

In Fig. A2 the Lindenberg measurements around the time of the Falcon overflight are shown together with a profile from WALES. Two separate panels are given since the DLR profiles used for the comparison with RAMSES and the sondes slightly differ, the balloon horizontally propagating along the flight path. The



comparisons are highly satisfactory. No systematic bias is found, and deviation clearly exceeding 5 % exist just in a few altitude ranges. In the intrusion layer the uncertainty of the WALES mixing ratio is 40 ppm (see lower panel of Fig. 7), i.e., much smaller than over Payerne due to less absorption. The RAMSES data are displayed for measurement times of 10 and 30 min. An improvement by the longer averaging is seen only above 8 km

where the noise of the 10-min data is high.

### *Zugspitze*

Figure A3 shows comparisons between the UFS 817-nm DIAL and the DLR 935-nm DIAL WALES. For this comparison we kept the smoothing interval of the UFS DIAL rather low, dynamically (nonlinearly) growing from about 25 m at 3 km to about 125 m at 10 km (definition: VDI, 1999). Three WALES profiles are given for

time intervals before, around and after the overflight of the mountain.

As one would expect from the co-ordinates the best agreement is found for the second WALES profile. In the altitudes ranges up to 4.5 km (e.g., moving spike at the concentration maximum) and between 5.3 and 7.3 km there is a considerable change in density along the flight path. At 6.6 km the $H_2O$ density is more than doubling between between 19:49 CET and 19:54 (see also colour change in Fig. 2), this time interval corresponding to a

flight distance of more than 70 km. The agreement of all three profiles is excellent up to 7 km. Above 7 km, the uncertainty for the ground-based system grows due to the considerable light absorption in the rather moist lower free troposphere.

The uncertainty of the Zugspitze DIAL in the dry layer is of the order of $1 \times 10^{21}$ m$^{-3}$, whereas a higher uncertainty (about $3.5 \times 10^{21}$ m$^{-3}$) is specified for the DLR system since 3 km a.s.l. means far-field detection for

the air-borne DIAL. Between 3.4 and 7.0 km an average difference between WALES and the UFS DIAL was determined as $8 \times 10^{20}$ m$^{-3}$ (1.6 % of the peak mixing ratio), the standard deviation of this value being $1.1 \times 10^{21}$ m$^{-3}$. The average difference is mainly determined by the offset in the range between 3.6 and 4.8 km which is in the far field of the DLR DIAL.

### Acknowledgements

The authors thank H. P. Schmid for his interest and support. The Zugspitze in-situ data were generated by H. E. Scheel who passed away in June 2013 after unfortunate surgery. W. Steinbrecht provided high-resolution radiosonde data of the German Weather Service for Stuttgart and Munich. The great support by the UFS team is acknowledged. The development of the Zugspitze water-vapour DIAL has been funded by the Bavarian Ministry of Economics and German Bundesministerium für Bildung und Forschung within the programme

Atmosphärenforschung 2000 (ATMOFAST project: Atmospheric Long-range Transport and its Impact on the Trace-gas Composition in the Free Troposphere over Central Europe, (ATMOFAST, 2005)). The observations of volcanic aerosol at Garmisch-Partenkirchen (UFS) contribute to NDACC (Network of the Detection of Atmospheric Composition Change) and EARLINET (European Aerosol Research Lidar Network, currently partly founded by ACTRIS 2).

The service charges for this open access publication have been covered by a Research Centre of the Helmholtz Association.



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



**Figures:**

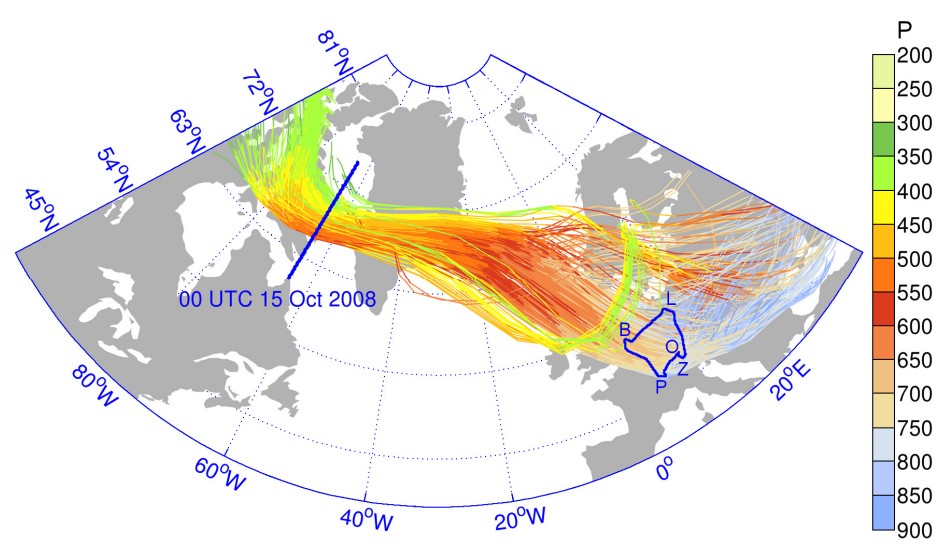

**Fig. 1.** LAGRANTO five-day forward trajectories from the full set calculated that fulfil both the deep-STT criterion and a passage above the blue line along the 65º-W meridian within 6 h of 15 October 2008, 0:00 UTC. The pressure level is colour coded. The blue contour around a major part of Central Europe visualizes the flight track of the DLR Falcon jet from Oberpfaffenhofen (O, southwest of Munich) to the four stations Payerne (P), Bilthoven (B), Lindenberg (L) and Zugspitze (Z), and back.





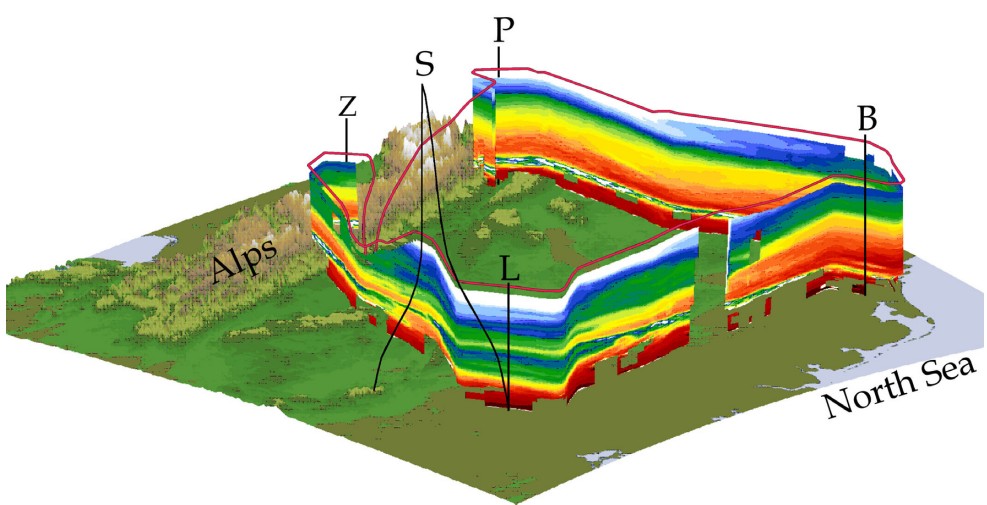

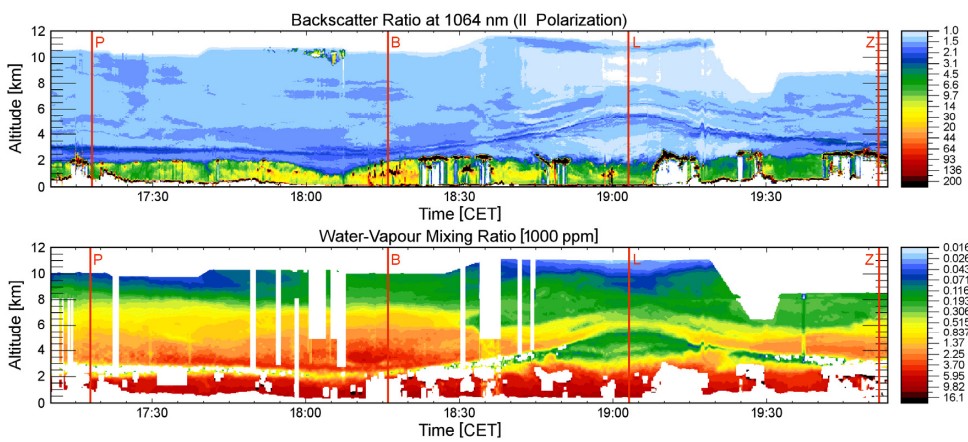

**Fig. 2.** . *Top panel:* Slant view from the north-east of the WALES flight in the late afternoon of 17 October 2008; P means Payerne (Switzerland), B Bilthoven (The Netherlands), L Lindenberg (North-East Germany), and Z Zugspitze (Garmisch-Partenkirchen, South Germany). The flight track (red line) is shown above the "curtain" of the water-vapour profiles from the air-borne DIAL measurements. The track of the Lindenberg sonde (S) is marked by a black line. *Bottom Panel:* Vertical distributions of water vapour (bottom) and 1064-nm aerosol backscatter ratio (top) along the flight track; the times of the overflights of the four stations are marked by red vertical lines.



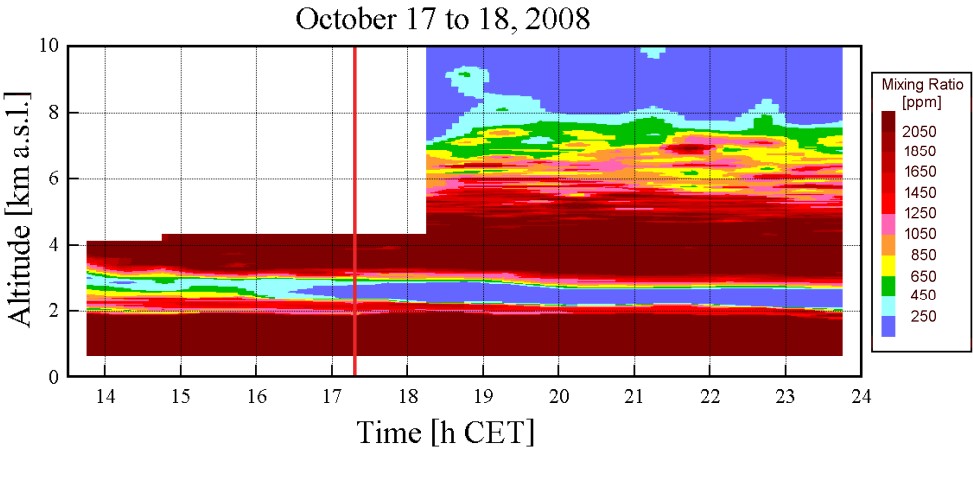

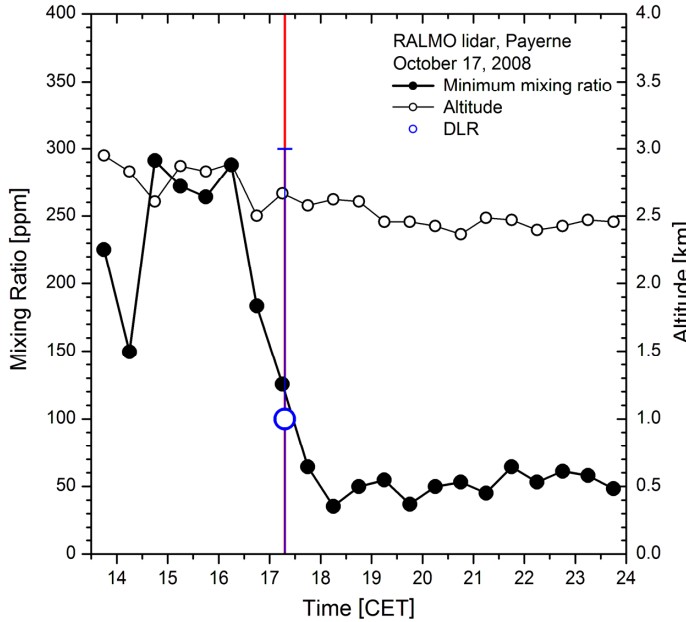

**Fig. 3.** *Upper panel:* Water-vapour time series of RALMO on 17 October 2008; the time of the aircraft overflight (17:18 CET) is marked by a red vertical line. The figure is based on 0.5-h averages, the times being centred in the respective measurement interval. Before 18:15 CET the residual daylight background prevented measurements beyond the lower free troposphere. *Lower panel:* Time series of the water-vapour minimum in the stratospheric intrusion layer on 17 October 2008, as recorded by RALMO, and the corresponding altitude; the time of the aircraft overflight (17:18 CET) is marked by a red vertical line. The figure is based on 0.5-h averages, the times being centred in the respective measurement interval.





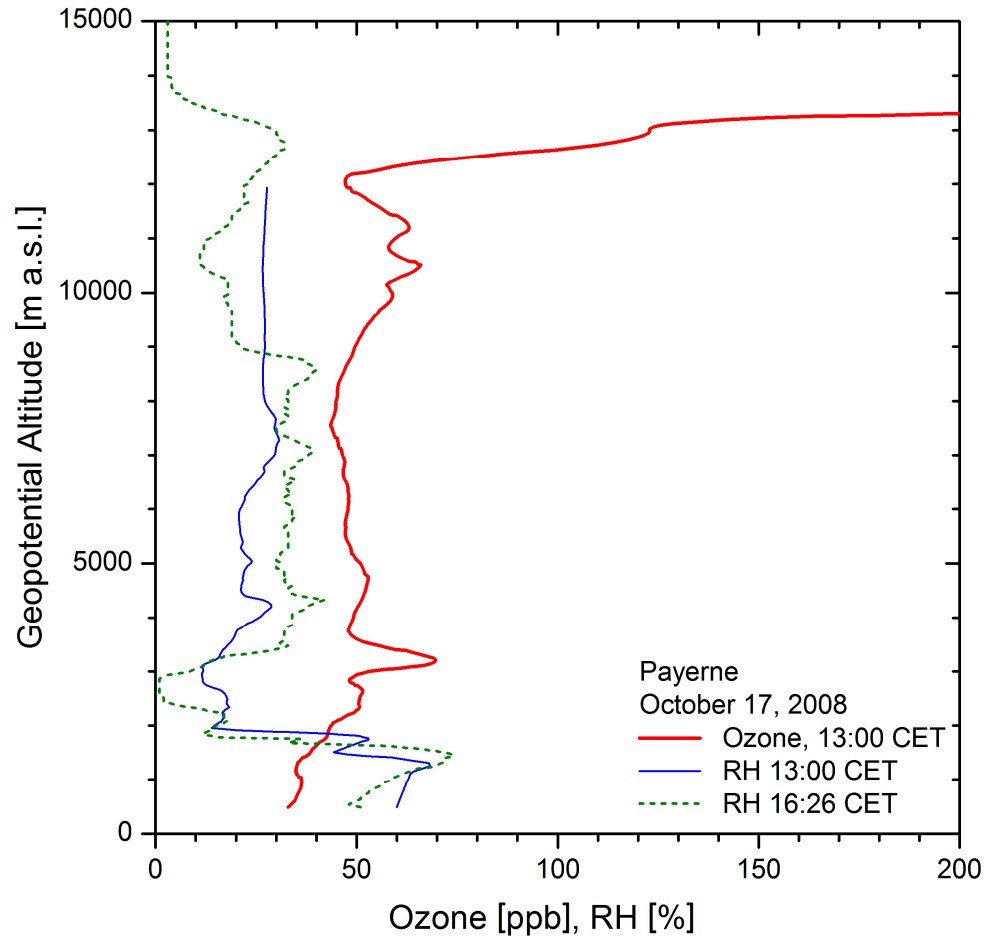

**Fig. 4.** Sonde ozone and relative-humidity profiles above Payerne on 17 October 2008; the times are launch times.





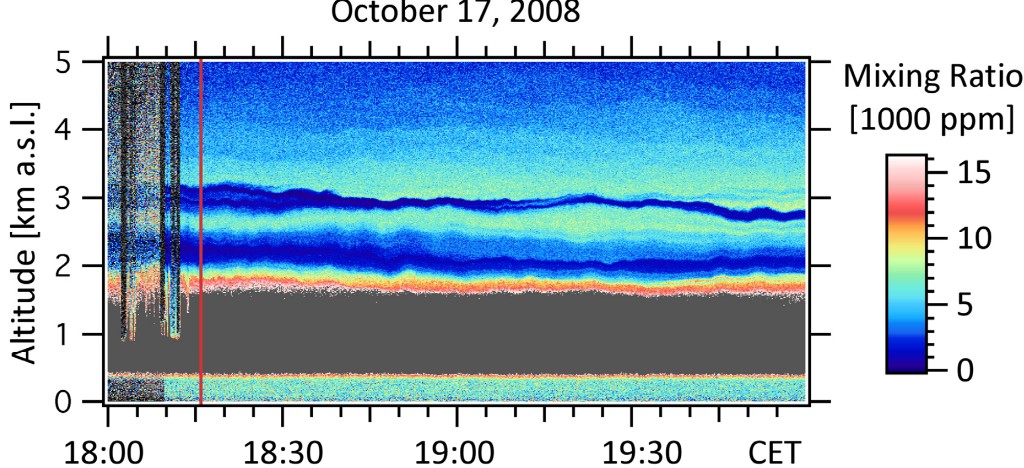

**Fig. 5.** Water-vapour time series of CAELI on 17 October 2008 (Bilthoven); the time of the aircraft overflight (18:16 CET) is marked by a red vertical line. In the graph, only the data from the far-field receiver are shown (above 1.7 km). The profiles are shown at the full native resolution of 10 s and 7.5 m.





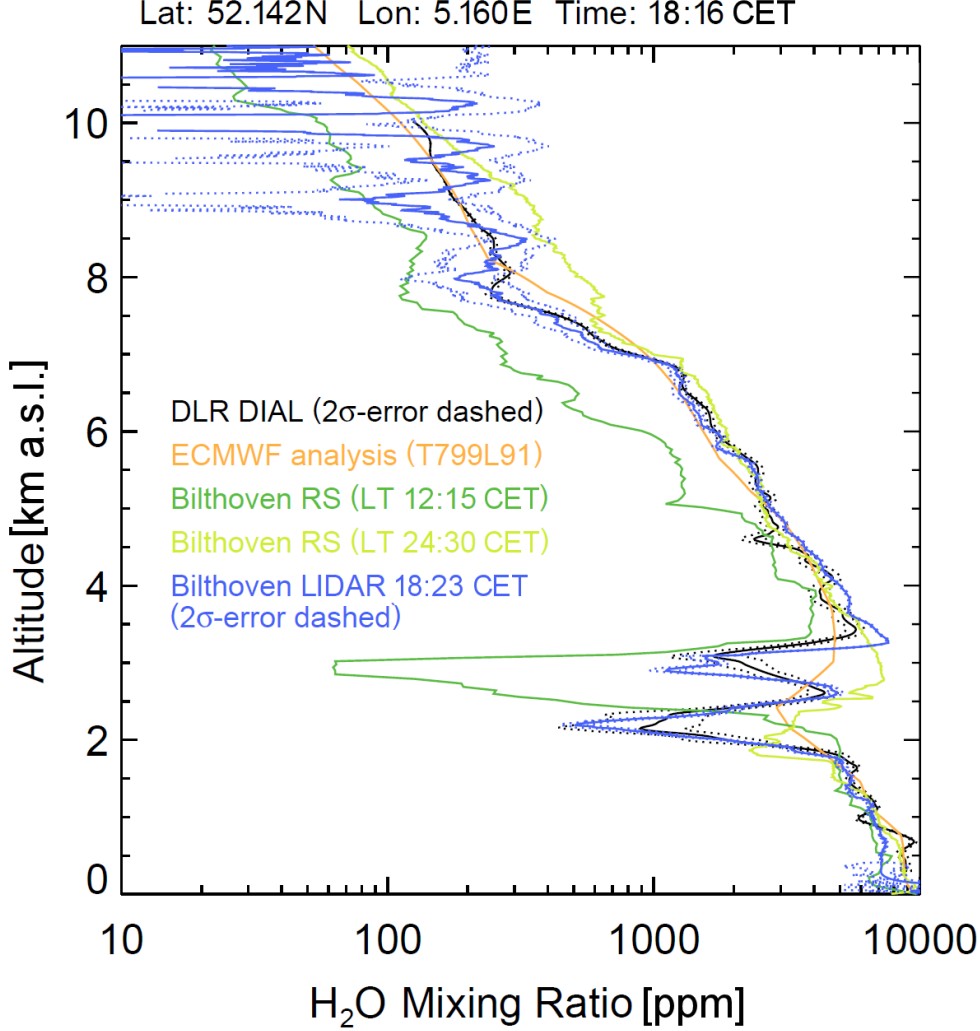

**Fig. 6.** Comparison of the water-vapour mixing ratio from the CAELI Raman lidar (18:23 CET, i.e., 15 min. average between 18:15 and 18:30)) and the air-borne lidar (18:15 CET) at Bilthoven; the mixing ratios from the routine noon and midnight measurement at De Bilt (station code 6260) are given for comparison. The noon profile reveals a much more pronounced stage of the intrusion than the lidar data. In addition, the humidity result from a high-resolution ECMWF analysis for the time of the aircraft arrival is shown, again just indicating the intrusion layers (one of the two). The times for the lidar systems refer to the middle of a measurement, for the sonde the launch time (LT) was taken.



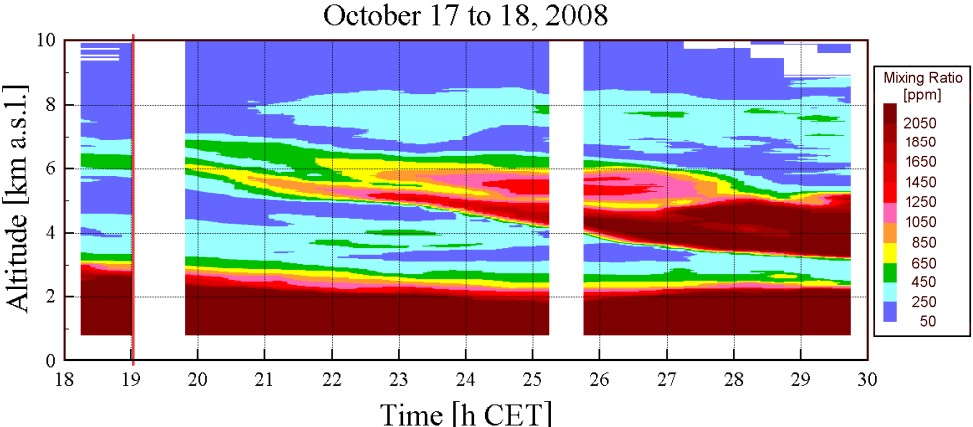

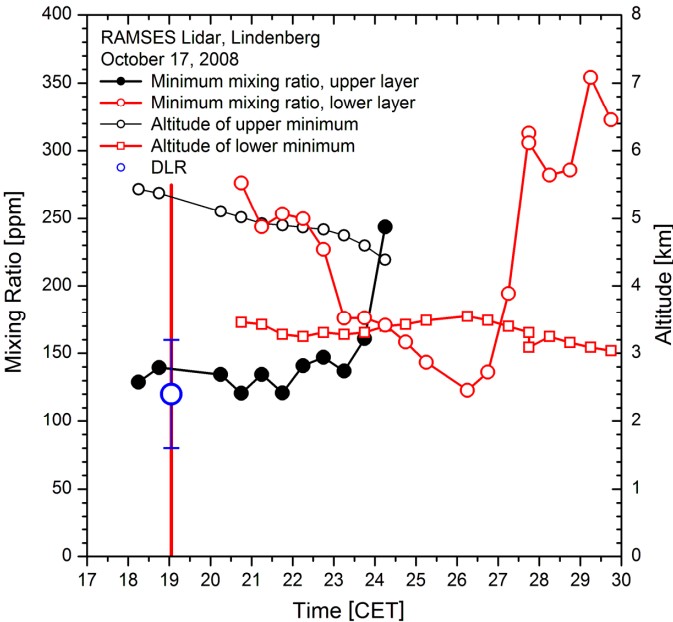

**Fig. 7.** *Upper panel*: Water-vapour time series of RAMSES on 17 October 2008 (Lindenberg); the time of the aircraft overflight (19:03 CET) is marked by a red vertical line. *Lower panel:* Time series of the two water-vapour minima in the stratospheric intrusion layer on 17 October 2008, as recorded by RAMSES, and the corresponding altitude; the time of the aircraft overflight (19:03 CET) is marked by a red vertical line. The figure is based on 0.5-h averages, the times being centred in the respective measurement interval.



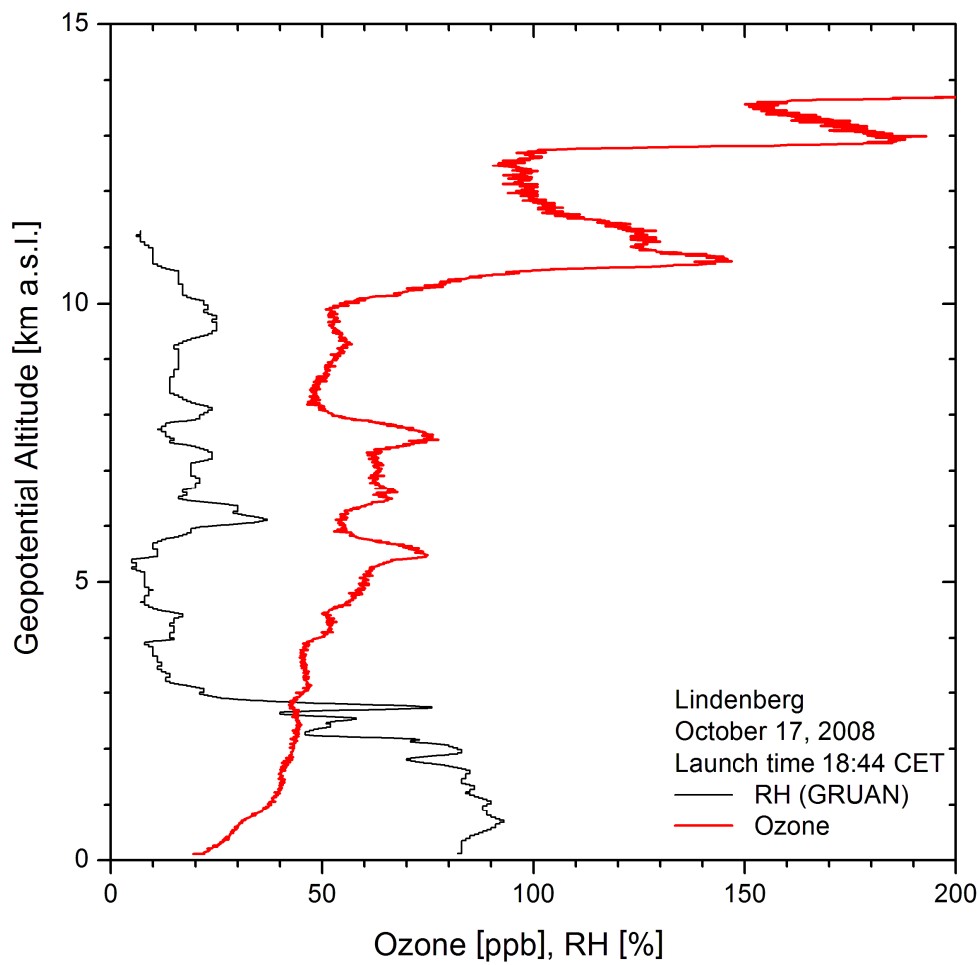

**Fig. 8.** Ozone and RH profiles during the sonde ascent launched at Lindenberg at 18:44 CET; for the RH of the RS92 sonde the final GRUAN data product was taken.




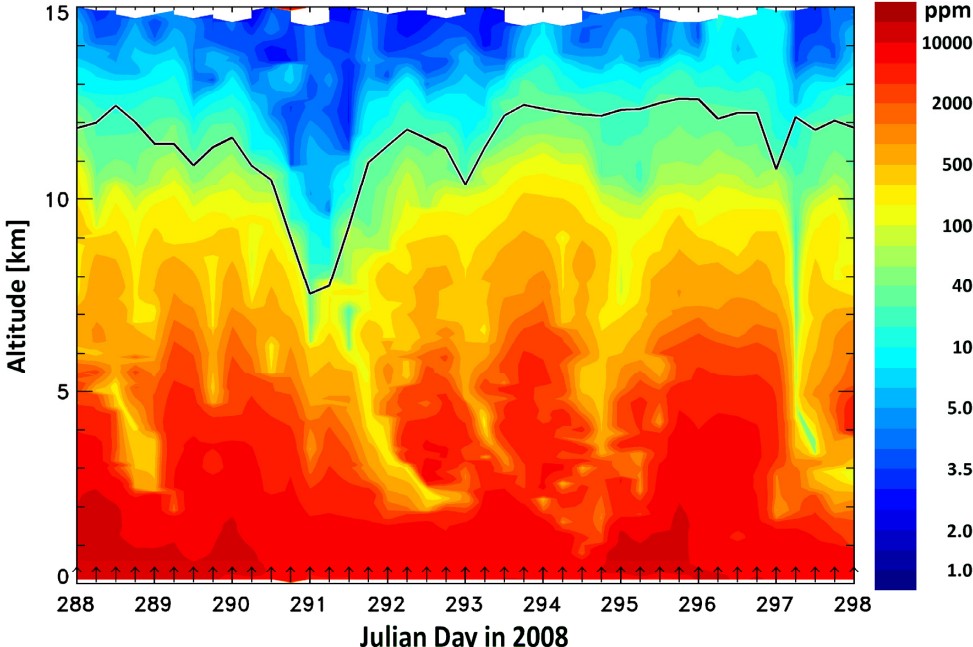

**Fig. 9.** Time series of the water-vapour mixing ratio over Lindenberg during the period between 14 (Julian day 288) and 23 (Julian day 297) October 2008; this figure was derived from radiosonde ascents at intervals of 6 h. The mean flight times (in UTC) are marked by arrows. The thermal tropopause is indicated by a black line. The intrusion examined in this study is visible on Julian days 291 (17 October) and 292. The graphics indicates the passage of a major part of the tropopause fold over Lindenberg.



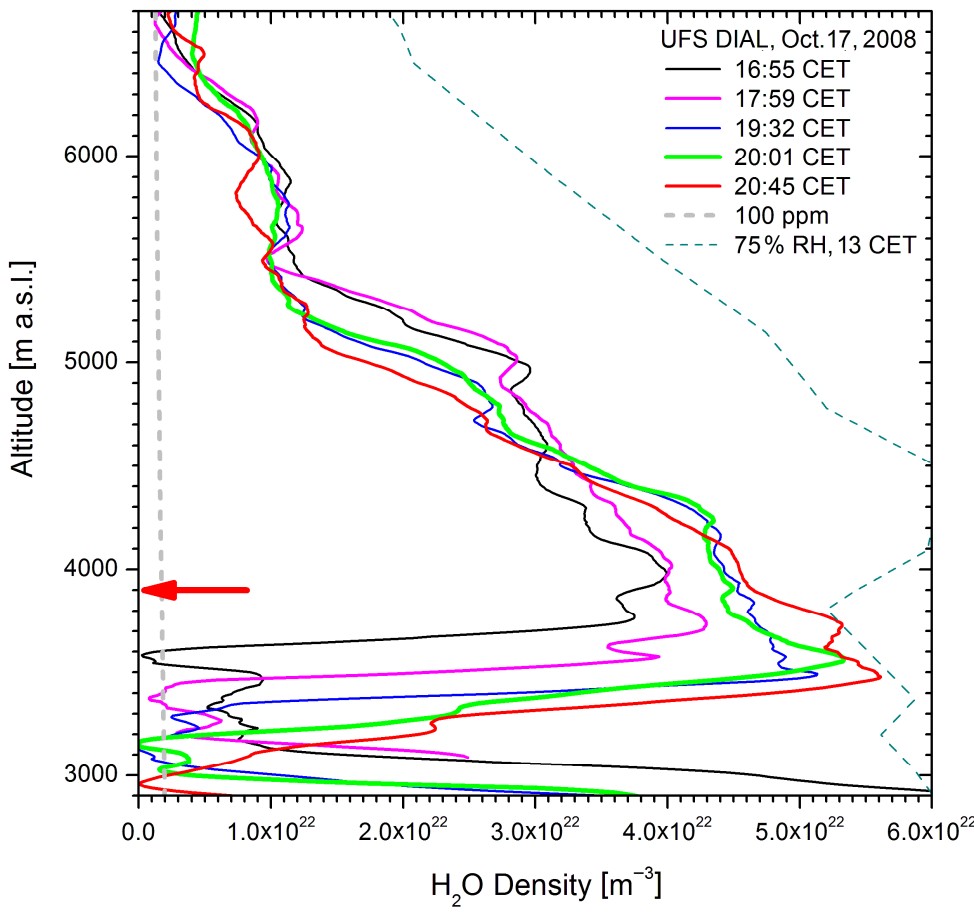

**Fig. 10.** Water-vapour profiles from the measurements of Zugspitze DIAL on 17 October 2008; the red arrow marks the vertical position of the RH minimum (1 %) from the noon "Munich" radiosonde, observed during the ascent at 11:56 CET. The grey dashed line marks a mixing ratio of 100 ppm as determined from the same sonde ascent. In addition, the density profiles for 75 % RH is given for a crude comparison (Munich, 13:00 CET). The corresponding profile for the following midnight shows significantly higher RH below 4.5 km since the intrusion had subsided to 2.78 km and, therefore, is not included here.





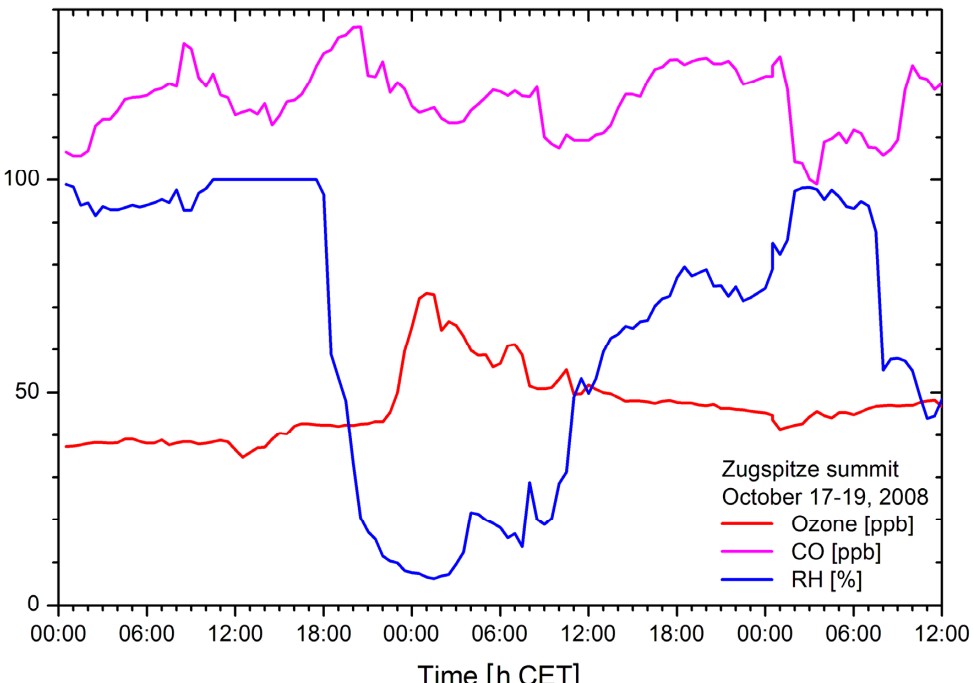

**Fig. 11.** Zugspitze in-situ measurements of ozone, carbon monoxide and relative humidity on 17-19 October 2008; the stratospheric layer is clearly visible in the $H_2O$ and $O_3$ data, but there is no significant hint in the CO curve.





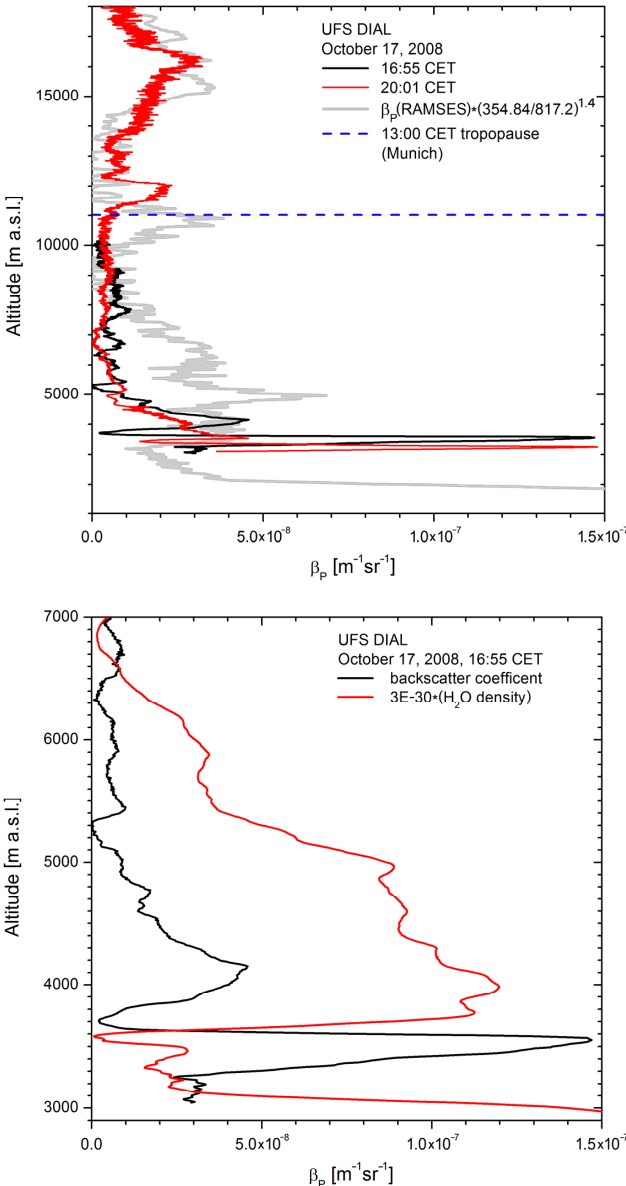

**Fig. 12.** *Upper panel:* 817.2-nm aerosol backscatter coefficients for two of the measurements at UFS on 17 October 2008; the two spikes between 3 and 4 km are located inside the intrusion layer at these times. For comparison a 354.84-nm profile from a 3-h average of RAMSES measurements around the time of the DLR overflight of Lindenberg is shown, rescaled for 817.2 nm. Here, the corresponding aerosol peak was detected at about 5 km. In the stratosphere two volcanic aerosol layers related to the Okmok and Kasatochi eruptions are seen. *Lower panel:* Vertically zoomed 16:55-CET measurements at UFS: 817.2-nm backscatter coefficients and water-vapour density, scaled to fit horizontally into the frame.





**Fig. 13.** *Right panels:* Intrusion trajectories (black lines) intersecting a vertical control surface (green line) above Northern Canada at three different times; *left panels:* intersection points of the trajectories on the control surface (see text); the blue contour lines are isentropes (in K).



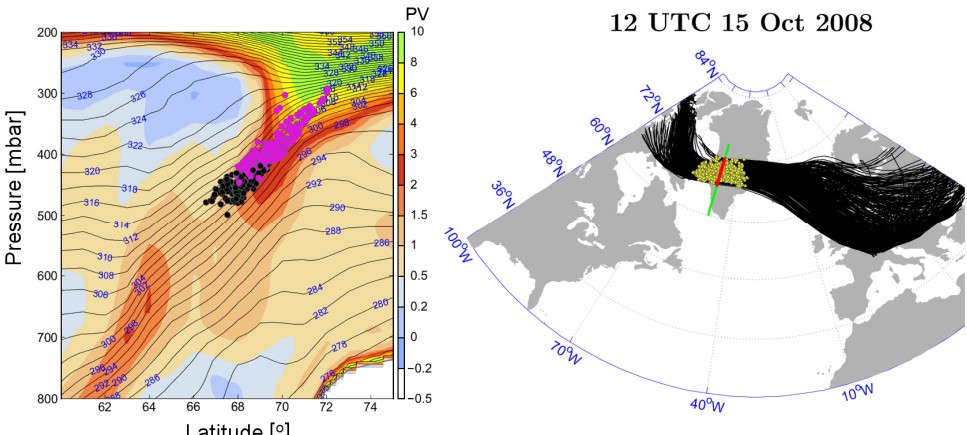

**Fig. 14.** *Right panel:* Intrusion trajectories (black lines) intersecting a vertical control surface (green line) above the west coast of Greenland at three different times; *left panel:* intersection points of the trajectories on the control surface (see text); the blue contour lines are isentropes (in K).





**Fig. 15.** *Right panels:* Intrusion trajectories (black lines) intersecting a vertical control surface (green line) along the coast of The Netherlands, Germany and Poland at three different times; *left panels:* intersection points of the trajectories on the control surface (see text); the blue contour lines are isentropes (in K).



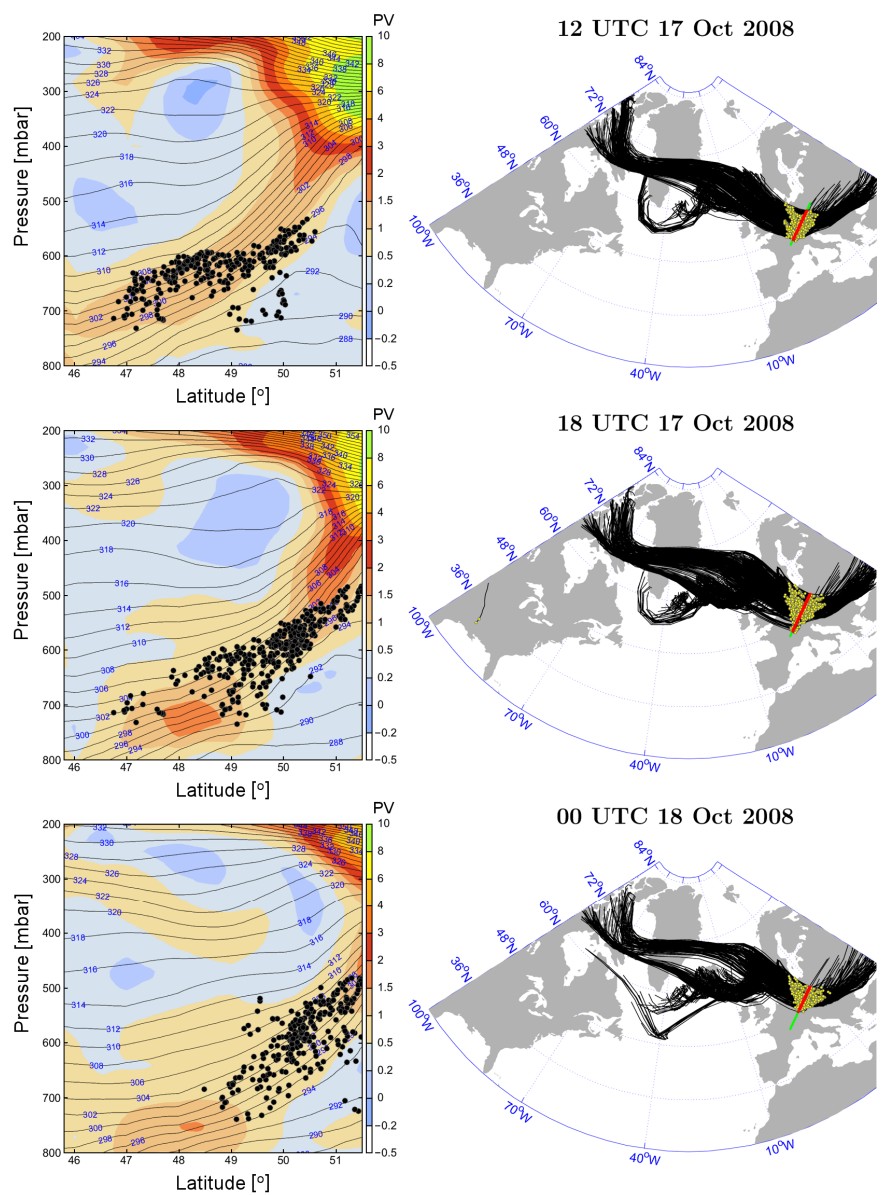

**Fig. 16.** *Right panels:* Intrusion trajectories (black lines) intersecting a vertical control surface (green line) north of the Alps at three different times; *left panels:* intersection points of the trajectories on the control surface (see text); the blue contour lines are isentropes (in K).



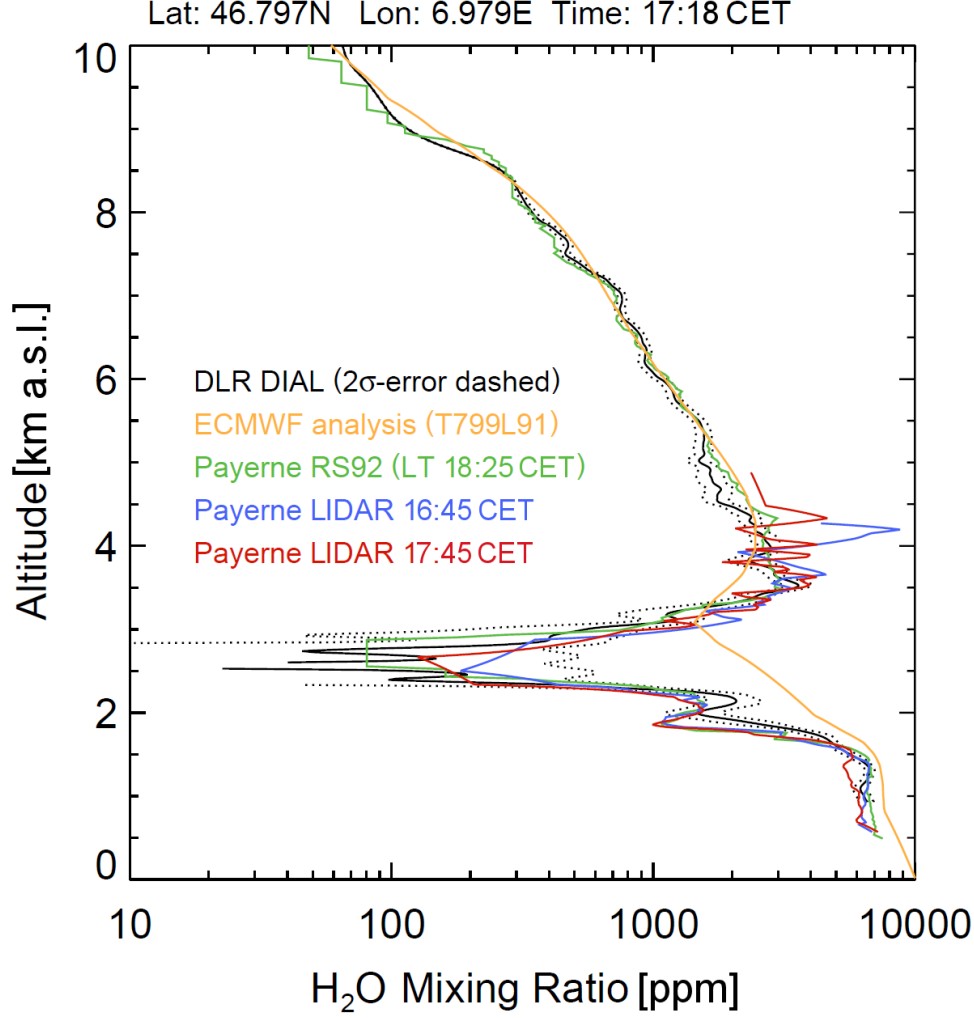

**Fig. A1.** Comparison of the water-vapour mixing ratio from the Raman and the air-borne lidar, and from a sonde ascent at Payerne; in addition, the corresponding humidity result from an ECMWF analysis is given which barely shows the intrusion layer. The times for the lidar systems (DLR: top of panel) refer to the middle of a measurement, for the sonde the launch time (LT) was taken.



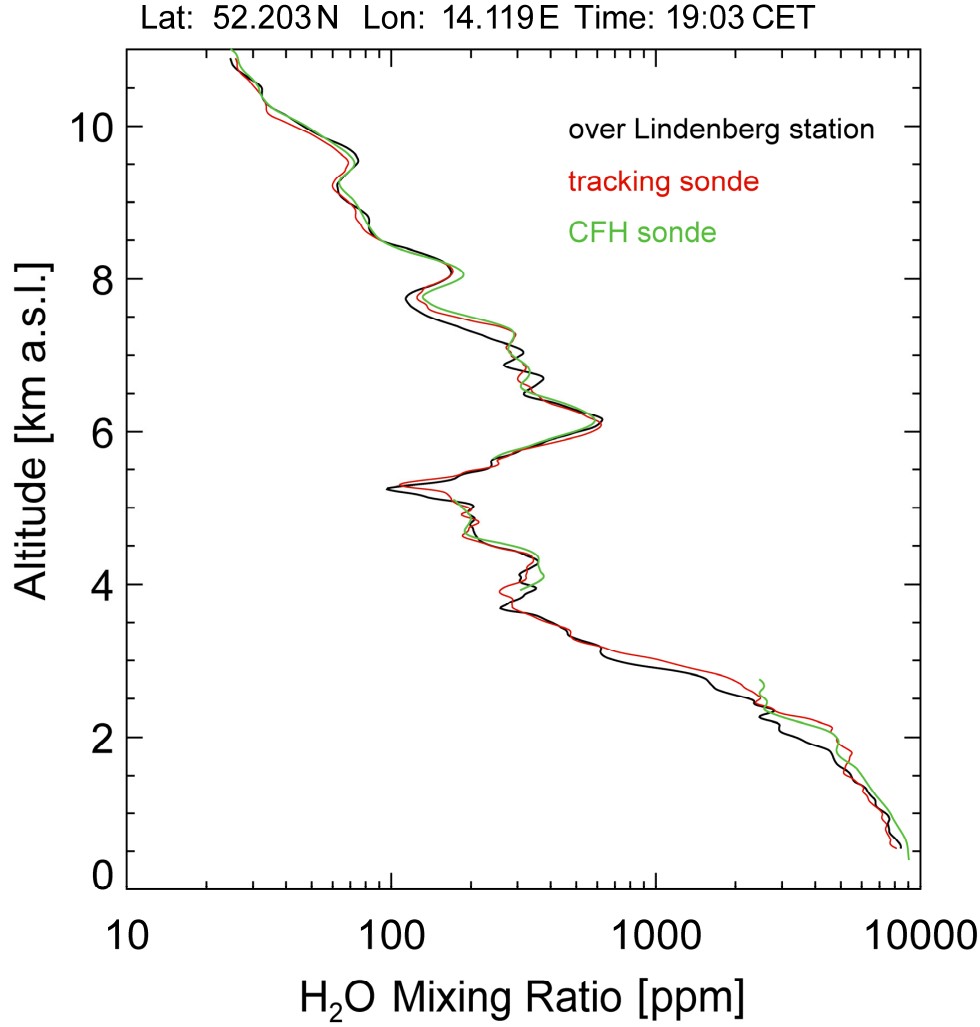

**Fig. A2a.** Upper panel: Comparison of WALES, a RS 92 and a CFH sonde over Lindenberg (Germany); the launch time of the balloon was 18:44 CET. The time given above the panel is that of the passage of the aircraft above the station. The WALES profile was evaluated slightly delayed for matching the average balloon position.



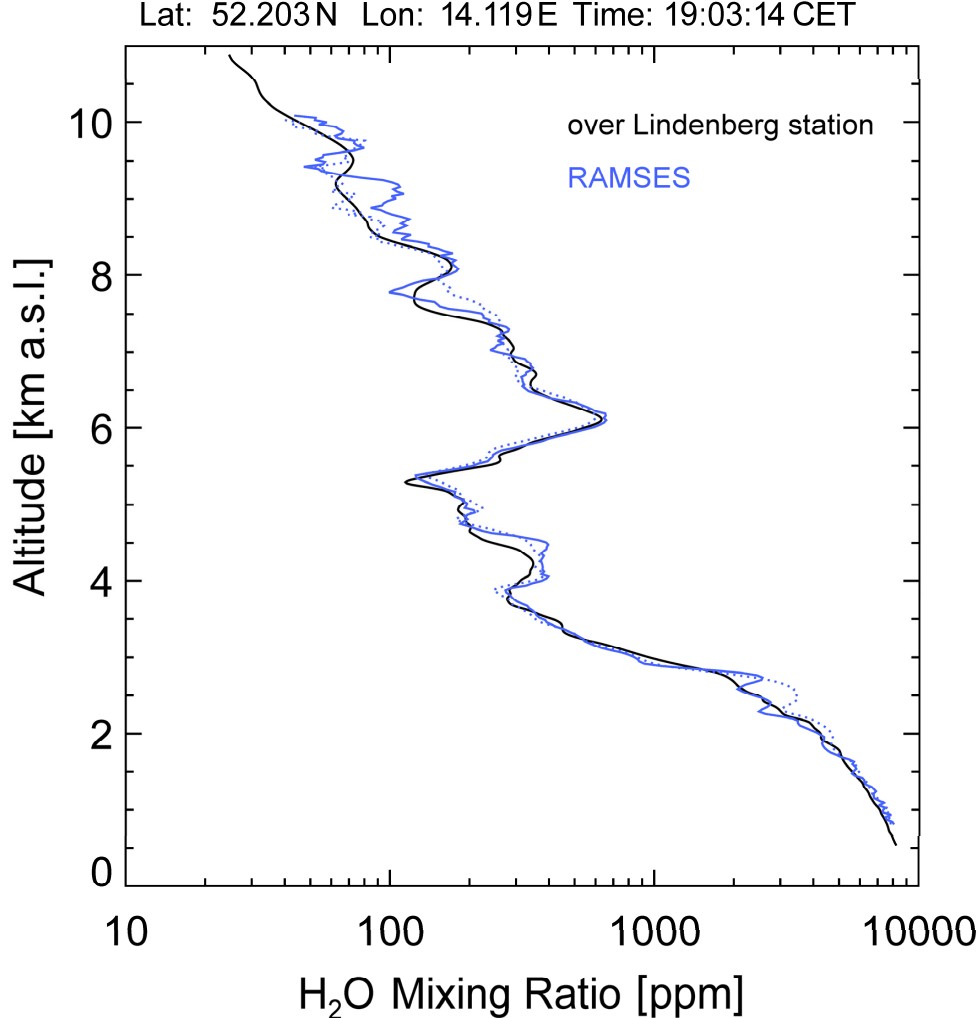

**Fig. A2b.** Lower panel: Comparison of the WALES an RAMSES lidar systems over Lindenberg (Germany); Lindenberg; for RAMSES two data accumulation times were taken, 10 min (blue line, central time 18:57 CET) and 30 min (broken blue line, central time 18:47 CET).




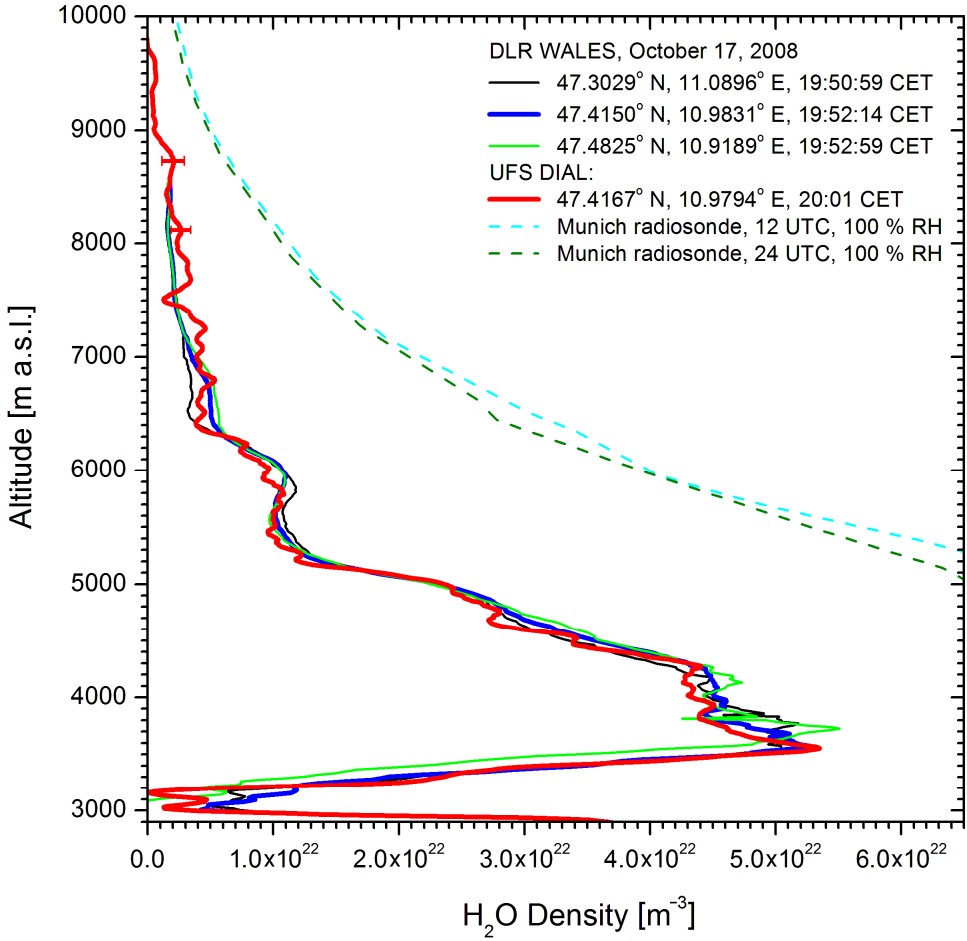

**Fig. A3.** Comparison of Zugspitze (UFS) DIAL and WALES; three WALES density profiles are shown around the time of the overflight. The best agreement was found for the best matching of the co-ordinates. The UFS data were smoothed less than in Fig. 10. The density profiles for 100 % RH is given in dashed lines (Munich, 1:00 and 25:00 CET).