# Peer review of "How stratospheric are deep stratospheric intrusions? - LUAMI 2008"

_Atmospheric Chemistry and Physics, 2016_

## Referee Comment (RC1) · Anonymous Referee #2 · 20 Apr 2016

The submitted manuscript provides a very nice insight in a case study of a deep stratospheric intrusion using water-vapour profiling measurements from high quality ground-based instruments at four observational sites in Central Europe (in particular the CFH sonde, differential-absorption and Raman lidar systems) as well as an air-borne lidar system creating a transect of humidity profiles between all four stations. The study is further supported by a thorough trajectory analysis of the event using LAGRANTO trajectory model. I suggest the manuscript can be accepted for publication after considering a few minor comments.

Comments 1) Introduction, Page 2, lines 5-7: There a few other studies of stratosphere to troposphere transport using lidar systems (e.g. Ancellet et al., JGR, 1991; Galani et al., JGR, 2003; Papayannis et al., Ann. Geopys., 2005). 2) Introduction: Since the study refers to a case study of a deep intrusion down to the lower troposphere, the

authors may add a few lines about the scarcity of these vigorous events and refer to similar cases studies that have been published in the literature indicating an influence even down to near surface. 3) Page 7, lines 11-12: Maybe you could provide rather more details for the methodology applied using LAGRANDO model. Furthermore when the authors state that " set of five-day trajectories was started in the entire region covering the Atlantic Ocean and Western Europe (20° E to 120° W and 40 to 80° N) between 200 and 600 hPa" do they mean that 5 forward trajectories were released at each grid point with 1x1 deg resolution within the domain from 20° E to 120° W and from 40 to 80° N for all model levels between 200 and 600 hPa? My estimate counts roughly at least 100000 trajectories. Hence it would be nice to provide some numbers of the total number of trajectories calculated and how many of these are selected as stratospheric intrusion trajectories. 4) Page 7, line 30: The authors state "in agreement with typical behaviour". Please describe what is a typical behaviour and provide the appropriate references. 5) Page 8, lines 11-15: Here, a bit more discussion on the enhanced aerosol content following Fig. 2 would be helpful for the reader. 6) page 10, line 34 as well as page 11, line 6: Does the time 17:02 CET refer to 16:55 CET shown in Figure 12? 7) page 11, lines 7-8: The authors state that the aerosol peak is located in the upper half of the intrusion layer. My impression from Figure 12 is that the aerosol layer covers the largest part of the dry intrusion layer. 8) Page 11, line 25: " ... fulfilling the deep-intrusion criteria ... " Are the deep-intrusion criteria those referred in Section 2.1.6 for the stratospheric intrusion trajectories? 9) Page 12: lines 25-26: The sentence " In any case ... of Fig.9" needs further elaboration as it is not clear enough. 10) Page 12, lines 32-36: This paragraph needs further elaboration as it is not clear enough. 11) Page 13, lines 1-2: There are also other recent modelling studies that share a similar perspective indicating that the role of stratosphere to troposphere transport to near surface background ozone may be of greater importance than previously anticipated in 1990s and 2000s (Lin et al., JGR 2012; Lefohn et al., Atmos. Environ., 2014

---

## Referee Comment (RC2) · Anonymous Referee #1 · 11 May 2016

General comments:

This paper presents a case study of a stratospheric intrusion observed over Germany by a set of instruments during the 2008 LUAMI field experiment. It is, in my opinion, one of most detailed stratospheric intrusion analysis I have ever seen. However, the question raised by the article's title "How stratospheric are deep stratospheric intrusions" is not answered. The paper falls short of giving convincing conclusions on how to proceed to estimate Stratosphere to Troposphere Transport (STT) flux of stratospheric air masses based on the observations used in the paper.

I'm not convinced by what this paper has to offer in terms of new findings or method to estimate STT flux. What knowledge do we gain from this case study compared to the existing literature on the subject, and Trickl et al. (2014) in particular, that is worth

being published in ACP? How can we extrapolate the results of this single event to other STT events in the mid-latitudes in general?

Even though this case study is well presented, I think that a lot of effort is still required to convince me that we may have a way to quantify the STT flux of this case study based on observation alone to make this article publishable in ACP.

Specific comments:

1) In the conclusion, the authors say "the results presented . . . are an important pre-requisite on the way to quantify STT based on observational data alone". From the conclusions on the observation and modeling efforts, I don't see how those results show any important prerequisite. Since no STT flux has been quantified, I still have doubt on our capability to estimate STT flux based on observations only. In fact, it is not possible unless the authors have a wind profiler that would provide the wind speed and direction of the stratospheric intrusion. Otherwise, the authors will need to use the 3D wind fields from a global model (e.g. ECMWF).

2) The conclusion on the observations is interesting. However, it doesn't help the reader to estimate how much information do we gain from each measurement tech-niques to subsequently quantify the STT flux of this case study. What do we need exactly to quantify the STT? Continuous LIDAR measurements from a ground site? LIDAR measurements on board an aircraft? High-resolution radiosondes? all of them? How does it potentially affect our capability to estimate a STT flux if one has access to part of those measurement techniques? A discussion is required.

3) Conclusion based on the modeling work: The trajectories show that the air mass comes from the vicinity of the tropopause. However, the authors didn't explain why the top of the intrusion layer had enhanced ozone and aerosol compared to the lower part. How is it possible that the lower part of the layer, which is supposed to be stratospheric, has no enhancement in ozone and aerosol?

The main hypothesis in the paper is that an air mass with 50ppmv of water vapor has a stratospheric origin. In Vaughan et al. (2005, ACP), it is shown (figure 5, a ozone/water vapor scatter plot) that dry air masses with 50ppmv of water vapor can be associated with low ozone. It means that an upper tropospheric air mass can have very low water vapor concentration in the mid-latitudes.

From the early 2000s (e.g. Cooper et al. 2002 and reference therein), we know that coherent air streams (in the Lagrangian sense) with distinct dynamic and chemical signatures exist in frontal systems. We know that the so called dry aistream, originating from the upper troposphere, is a downward propagating air mass behind the cold front with a dry air signature. The dry air stream is clearly visible in water vapor satellite images. It is, in general, associated with a tropopause fold with a stratospheric signature.

The dry layer studied in this paper, considered to be of stratospheric origin, is probably the dry airstream of a frontal system. In this case, it makes sense that the stratospheric intrusion will be located at the top of the dry airstream as both upper tropospheric and stratospheric air masses will move downward and will follow the same vertical layering.

Therefore, the dry signature of a layer cannot be used to define the stratospheric origin of an air mass in general, which contradicts the main results of this paper. Can the authors provide more evidence that the water vapor concentration in the upper troposphere should be higher than 50ppmv?

It would be good if the authors show water vapor channel images from satellites to see exactly the synoptic situation of this event.

Reference: Cooper, O. R., J. L. Moody, D. D. Parrish, M. Trainer, T. B. Ryerson, J. S. Holloway, G. Hübler, F. C. Fehsenfeld, and M. J. Evans (2002), Trace gas composition of midlatitude cyclones over the western North Atlantic Ocean: A conceptual model, J. Geophys. Res., 107(D7), 4056, doi:10.1029/2001JD000901.

4) The justification of the paper is to provide some elements to estimate a STT flux

based on observations alone. A discussion is missing on the limitation of STT flux estimates. The authors showed that the stratospheric ozone was originating from a layer right above the tropopause, within the mixing layer around the tropopause region. Such an air mass has a short residence time in the stratosphere, and cannot be considered as pure stratospheric. How does that affect a STT flux estimate?

Using 5-day trajectories, such an air mass might be defined as stratospheric. However, a Lagrangian particle dispersion model would show that in fact the chemical composition of the air mass results from the successive mixing of tropospheric and stratospheric air masses in the vicinity of the tropopause. Such a mixing will depend on the duration of the Lagrangian trajectories. How does the author justify the use of 5-day trajectories instead of 10 days? How sensitive the results would be to the definition of the tropopause used?

6) The title of the article is misleading because a single stratospheric intrusion is analyzed in this study. I don't think a case study can help answer such question.

Technical comments:

Page 2, line 29: you mean in stratosphere in the mid-latitudes. Page 8, line 14: "around the middle of 2008": can you be more specific? Page 10, lines 23-29: the CO is reduced by 20ppb in the so-called stratospheric layer. I wouldn't say that "CO did not change much".

---

## Author Comment (AC1) · 11 Jun 2016

**Replies to the two reports on**

**"How stratospheric are deep stratospheric intrusions? − LUAMI 2008"**

by Thomas Trickl et al.

I thank both reviewers for the thoughtful comments. However, quite a few of the remarks contain goals that are beyond the scope of this paper. These issues are planned to be tackled in a subsequent publication.

The comments are shown in italics, the answers normal. The revised manuscript is submitted parallel with the changes marked in red colour.

**Anonymous Referee #1**

*General comments:*

*This paper presents a case study of a stratospheric intrusion observed over Germany by a set of instruments during the 2008 LUAMI field experiment. It is, in my opinion, one of most detailed stratospheric intrusion analysis I have ever seen. However, the question raised by the article's title "How stratospheric are deep stratospheric intrusions" is not answered. The paper falls short of giving convincing conclusions on how to proceed to estimate Stratosphere to Troposphere Transport (STT) flux of stratospheric air masses based on the observations used in the paper. I'm not convinced by what this paper has to offer in terms of new findings or method to estimate STT flux. What knowledge do we gain from this case study compared to the existing literature on the subject, and Trickl et al. (2014) in particular, that is worth being published in ACP? How can we extrapolate the results of this single event to other STT events in the mid-latitudes in general?*

*Even though this case study is well presented, I think that a lot of effort is still required to convince me that we may have a way to quantify the STT flux of this case study based on observation alone to make this article publishable in ACP. Specific comments:*

This paper is Part 2 of a very detailed study on the topic specified in the title published in 2014. There, more information is given. This second part was needed because the experimental complexity of the LUAMI campaign required extra space. To our opinion publishing the results of the LUAMI campaign has its own right. This case has allowed us to characterize a rather homogeneous intrusion layer during the entire descent 1. I am not aware of studies yielding a similarly complete tomography of an intrusion event. Such a study was not possible for the cases described in Part 1 because of filamentation, and it finally yields a confirmation of what had been concluded from observations.

The two papers clearly address the question of their common title. It is found that even for rather long descent the concentrations change much less than thought earlier and that the input from the (UT)LS has a mixture of both stratospheric and tropospheric nature. The tropospheric contribution changes from case to case. We now find a confirmation of the hypothesized rather shallow outflow from just above the tropopause. We show that even the transition from a more tropospheric lower boundary to a more stratospheric nature towards the layer top can be seen under certain conditions.

A separate paper on the STT budget derived from the Zugspitze time has been planned for about a decade (based on the revised methods described in our 2010 paper) and will soon be started based on the data of the late Hans-Eckhart Scheel. This is now mentioned in the "Discussion" section. For this paper several approaches for estimating the mixed source contribution will be tested.

1) *In the conclusion, the authors say "the results presented : : : are an important prerequisite on the way to quantify STT based on observational data alone". From the conclusions on the observation and modeling efforts, I don't see how those results show any important prerequisite. Since no STT flux has been quantified, I still have doubt on our capability to estimate STT flux based on observations only. In fact, it is not possible unless the authors have a*

*wind profiler that would provide the wind speed and direction of the stratospheric intrusion. Otherwise, the authors will need to use the 3D wind fields from a global model (e.g. ECMWF).*

As indicated above, I added two sentences regarding the planned follow-up study for clarification. Some more discussion was added to the Zugspitze chapter in the "Results" section. The methods to be used have been described in several papers, in particular by Trickl et al. (2010). Thus, I had tried to be too short here. The outcome will be annual ozone fractions (like in (Beekmann et al., 1997; Stohl et al., 2000), and not fluxes. The fractions will be derived by data filtering, by correlating ozone with RH and [7]Be. The key finding of the previous and present studies for that effort is that mixing with tropospheric air during the descent can be almost neglected. With stronger mixing there would be the need to quantify the mixing, a very difficult task (Trickl et al., 2010; 2014).

*2) The conclusion on the observations is interesting. However, it doesn't help the reader to estimate how much information do we gain from each measurement techniques to subsequently quantify the STT flux of this case study. What do we need exactly to quantify the STT? Continuous LIDAR measurements from a ground site? LIDAR measurements on board an aircraft? High-resolution radiosondes? all of them? How does it potentially affect our capability to estimate a STT flux if one has access to part of those measurement techniques? A discussion is required.*

The methods were derived in our 2010 paper, which is now mentioned. I plan to focus on the Zugspitze results because of the possibility of correlating the data (data filtering). The lidar measurements have helped us in analysing some of the necessary details. In addition (as indicated in "Part I"), they have also shown that the fraction of intrusion days (free troposphere) in our measurement days can be very high. I meanwhile found a fraction of 80% for several years since 2007 in the entire free troposphere.

*3) Conclusion based on the modeling work: The trajectories show that the air mass comes from the vicinity of the tropopause. However, the authors didn't explain why the top of the intrusion layer had enhanced ozone and aerosol compared to the lower part. How is it possible that the lower part of the layer, which is supposed to be stratospheric, has no enhancement in ozone and aerosol?*

I do not agree: We do give an explanation both in the "Results" and the "Discussion" (P. 15, starting at line 35) sections. Nevertheless, I made some adjustments. Not the lower part is more stratospheric: it is the upper part! As visualized by the model calculations there is no inversion of the vertical distribution! The outflow from the stratosphere occurs perpendicularly to the fold.

The tropopause region is a transition region where ozone does not abruptly jump from tropospheric to fully stratospheric values. It is a mixture of both parts of the atmosphere. Years of observations show that most of the layer volume is filled with elevated ozone. What we see here is a rise in concentration (of both ozone and aerosol) with altitude, as one would expect for the source region. Maybe this unperturbed transfer of the vertical distribution cannot be generalized, but it is nice to see it here.

In the past the ozone and the $H_2O$ DIAL measurements in the Garmisch-Partenkirchen area had not been fully synchronized and there is also some orographic vertical shift of the distributions between both lidar sites. Thus, such a comparison could not easily be made. The $H_2O$ and aerosol measurement presented in Fig. 12b were both made with the $H_2O$ DIAL.

*The main hypothesis in the paper is that an air mass with 50ppmv of water vapor has a stratospheric origin. In Vaughan et al. (2005, ACP), it is shown (figure 5, a ozone/water vapor scatter plot) that dry air masses with 50ppmv of water vapor can be associated with low ozone. It means that an upper tropospheric air mass can have very low water vapor concentration in the mid-latitudes.*

The main result is that mixing with tropospheric air is much slower than expected from previous work. In "Part 1" we discussed all this in some more detail. We have observed 50 ppm, but also

much less (even here with the Zugspitze DIAL). 50 ppm could be estimated in "Part 1" to be in agreement with the 50 % of tropospheric contribution of the UTLS air in at least two papers. We hypothesize that this contribution comes from rising air streams that arrived at the tropopause before the beginning of the intrusion. The conclusion was: If the humidity was not picked up in the tropopause region this cannot occur much later because of a rapid increase of humidity just below the tropopause region.

Of course, there are some exclusive UT contributions, which we have seen over the years. In most cases, they can be discriminated. However, cases with no increase in ozone at all are rare. The increase may be small if the intrusion layer originates in the lowest part of the ozone rise (see last figure of our 2014 paper).

*From the early 2000s (e.g. Cooper et al. 2002 and reference therein), we know that coherent air streams (in the Lagrangian sense) with distinct dynamic and chemical signatures exist in frontal systems. We know that the so called dry airstream, originating from the upper troposphere, is a downward propagating air mass behind the cold front with a dry air signature. The dry air stream is clearly visible in water vapor satellite images.*

*It is, in general, associated with a tropopause fold with a stratospheric signature. The dry layer studied in this paper, considered to be of stratospheric origin, is probably the dry airstream of a frontal system. In this case, it makes sense that the stratospheric intrusion will be located at the top of the dry airstream as both upper tropospheric and stratospheric air masses will move downward and will follow the same vertical layering. Therefore, the dry signature of a layer cannot be used to define the stratospheric origin of an air mass in general, which contradicts the main results of this paper. Can the authors provide more evidence that the water vapor concentration in the upper troposphere should be higher than 50ppmv?*

*It would be good if the authors show water vapor channel images from satellites to see exactly the synoptic situation of this event.*

*Reference: Cooper, O. R., J. L. Moody, D. D. Parrish, M. Trainer, T. B. Ryerson, J. S. Holloway, G. Hübler, F. C. Fehsenfeld, and M. J. Evans (2002), Trace gas composition of midlatitude cyclones over the western North Atlantic Ocean: A conceptual model, J.Geophys. Res., 107(D7), 4056, doi:10.1029/2001JD000901.*

Thank you for that reference! I know it and I have used such images previously.

I inspected METEOSAT water-vapour images (GOES East does not cover well the regions of interest). They show some moderate drying after the frontal passage preceding the event. An indication of the intrusion over Central Europe is just seen in its northern part, i.e., where the layer was located at higher altitude. Further to the south no structure is visible, perhaps because the thin layer dives into the moist background. The contrast of the METEOSAT images is low. Given the high number of figures already present I, therefore, decided not to show one of these images. However, I added a few sentence to the first chapter of the "Results" section.

*4) The justification of the paper is to provide some elements to estimate a STT flux based on observations alone. A discussion is missing on the limitation of STT flux estimates. The authors showed that the stratospheric ozone was originating from a layer right above the tropopause, within the mixing layer around the tropopause region. Such an air mass has a short residence time in the stratosphere, and cannot be considered as pure stratospheric. How does that affect a STT flux estimate?*

Let us say: this has become an important motivation. Of course, all started by the surprise to see again and again this very low humidity values. Until we gradually started to realize this, the mixing had been a serious issue in the interpretation of the data filtering efforts of H. E. Scheel since the in-situ RH was rarely below 10 %. Another motivation for this study has been the unique qualityassured data set obtained from LUAMI that has allowed us to evaluate quite a few more details of this "textbook case".

As mentioned: we do not aim at deriving fluxes! The fact that the transition layer in the UTLS is not purely stratospheric is an important issue. A number of papers mention a tropospheric fraction of the order of 50 %. Any assessment of STT must include a treatment of this fact, and I plan to make a sensitivity study. We discussed all this in "Part 1" and I do not want to repeat the entire discussion. Was hier schreiben????

*Using 5-day trajectories, such an air mass might be defined as stratospheric. However, a Lagrangian particle dispersion model would show that in fact the chemical composition of the air mass results from the successive mixing of tropospheric and stratospheric air masses in the vicinity of the tropopause. Such a mixing will depend on the duration of the Lagrangian trajectories. How does the author justify the use of 5-day trajectories instead of 10 days? How sensitive the results would be to the definition of the tropopause used?*

As replied above mixing should be mostly limited to the tropopause region. Otherwise, the humidity would increase rapidly even during the first part of the descent. This is concluded from the observations alone (Part 1). Dispersion models overestimate mixing and are not used here (see part 1 for two FLEXPART simulations). The LAGRANTO model used here has almost always correctly forecasted intrusions for more than 15 years now (Trickl et al., 2010).

*6) The title of the article is misleading because a single stratospheric intrusion is analyzed in this study. I don't think a case study can help answer such question.*

The title refers to both papers (we now explicitly mention this in the Introduction). A total of about eighty cases has been studied! I added this information to the Introduction. Here, we treat a less complex case (less filamentary), but this time in some more detail and in a greatly extended observational range.

**Technical comments:**

*Page 2, line 29: you mean in stratosphere in the mid-latitudes.*

Clarified! We have examined sonde data next to our station and in the source regions of the individual intrusions (2014 paper).

*Page 8, line 14: "around the middle of 2008": can you be more specific?*

The eruptions occurred over extended periods of time. I added the first days of eruption for which, indeed, the highest altitudes were found.

*Page 10, lines 23-29: the CO is reduced by 20ppb in the so-called stratospheric layer. I wouldn't say that "CO did not change much".*

Changed!

**Anonymous Referee #2**

*The submitted manuscript provides a very nice insight in a case study of a deep stratospheric intrusion using water-vapour profiling measurements from high quality groundbased instruments at four observational sites in Central Europe (in particular the CFH sonde, differential-absorption and Raman lidar systems) as well as an air-borne lidar system creating a transect of humidity profiles between all four stations. The study is further supported by a thorough trajectory analysis of the event using LAGRANTO trajectory model. I suggest the manuscript can be accepted for publication after considering a few minor comments.*

*Comments*

*1) Introduction, Page 2, lines 5-7: There a few other studies of stratosphere to troposphere transport using lidar systems (e.g. Ancellet et al., JGR, 1991; Galani et al., JGR, 2003; Papayannis et al., Ann. Geopys., 2005).*

There are not only a few more lidar studies! The papers suggested here are all known to me and have been cited by me in the past. Here, I had selected a few lidar publications presenting time series. I apologize for having forgotten to include the classical paper by Ancellet et al. (1991) which is now added. I also added Galani et al.: Instead of a time series a larger number of cases are presented. This extension of the list necessitates to add a few more American lidar papers in order to make the European contribution less dominant.

*2) Introduction: Since the study refers to a case study of a deep intrusion down to the lower troposphere, the authors may add a few lines about the scarcity of these vigorous events and refer to similar cases studies that have been published in the literature indicating an influence even down to near surface.*

The impact of STT on the lowermost atmosphere is a topic that has gained a lot of interest in recent years, as one can see from the numerous papers published. Penetration into the PBL is rare in our record. However, in our new paper we do not tackle this issue. We concentrate on intrusions reaching the lower free troposphere and are not affected by strong convective mixing. These events are not scarce at all. 3000 m is reached many times per month in winter (Trickl et al., 2010).

*3) Page 7, lines 11-12: Maybe you could provide rather more details for the methodology applied using LAGRANDO model. Furthermore when the authors state that " set of five-day trajectories was started in the entire region covering the Atlantic Ocean and Western Europe (20 E to 120 W and 40 to 80N) between 200 and 600 hPa" do they mean that 5 forward trajectories were released at each grid point with 1x1 deg resolution within the domain from 20 E to 120 W and from 40 to 80 N for all model levels between 200 and 600 hPa? My estimate counts roughly at least 100000 trajectories. Hence it would be nice to provide some numbers of the total number of trajectories calculated and how many of these are selected as stratospheric intrusion trajectories.*

A few more details are added. "five-day trajectory" does not mean "five trajectories" which is made clear by using the hyphen! However, I clarifies this somewhat more.

*4) Page 7, line 30: The authors state "in agreement with typical behaviour". Please describe what is a typical behaviour and provide the appropriate references.*

I changed the phrase to "in agreement with the well-known transverse typical fold structure (e.g., Danielsen, 1968)".

*5) Page 8, lines 11-15: Here, a bit more discussion on the enhanced aerosol content following Fig. 2 would be helpful for the reader.*

An observation of aerosol in intrusions has been rarely reported. This could be due to the long background phase of the stratospheric aerosol and perhaps less attention in early ozone lidar measurements. This is discussed in more detail in the Discussion section, and examples are given in the chapters on ground-based measurements. I added a sentence making this clear.

*6) page 10, line 34 as well as page 11, line 6: Does the time 17:02 CET refer to 16:55 CET shown in Figure 12?*

16:55 CET is correct: 17:02 CET is the end time of the measurement, but we finally decided to specify the central time in this paper. Thank you for reading that carefully!

*7) page 11, lines 7-8: The authors state that the aerosol peak is located in the upper half of the intrusion layer. My impression from Figure 12 is that the aerosol layer covers the largest part of the dry intrusion layer.*

Thank you for this remark! Indeed, aerosol is found in a major part of the dry layer. "Peak" is a somewhat ambiguous word and I made a slight adjustment. However, the aerosol clearly rises from the bottom of the intrusion layer towards the top. Here, also the humidity reaches its minimum.

*8) Page 11, line 25: " ... fulfilling the deep-intrusion criteria ... " Are the deep-intrusion criteria those referred in Section 2.1.6 for the stratospheric intrusion trajectories?*

You certainly mean Sec,. 2.2! Yes, now specified!

*9) Page 12: lines 25-26: The sentence " In any case ... of Fig.9" needs further elaboration as it is not clear enough.*

I added the sentence "Thus, we cannot judge if the layer starts to detach from the fold." for a better connection. This topic has puzzled me since A. Stohl´s simulation in our 1999 paper showed a separation of the layer prior to the beginning of the lidar measurements.

*10) Page 12, lines 32-36: This paragraph needs further elaboration as it is not clear enough.*

I tried to improve this paragraph.

*11) Page 13, lines 1-2: There are also other recent modelling studies that share a similar perspective indicating that the role of stratosphere to troposphere transport to near surface background ozone may be of greater importance than previously anticipated in 1990s and 2000s (Lin et al., JGR 2012; Lefohn et al., Atmos. Environ., 2014*

As said there are even many more recent papers on this topic. The most recent one, Ott et al., JGR 2016, p. 3687, is in strong agreement with our own findings that a penetration of an intrusion layer the PBL is rarely observed. We have seen intrusion layers sliding along the top of the PBL at about 3000 m for more than a day. In the morning the Zugspitze station saw the dry air. Then, with the expansion of the PBL, the signatures of the layer disappeared, but returned in the evening.

However, we do not discuss the near-surface issues here. This would be a separate study requiring combined RH + $^7$Be analyses that are beyond the scope of the current paper. Dry layers are sometimes seen in the PBL in radiosonde profiles!